# Paracrine Met signaling triggers epithelial–mesenchymal transition in mammary luminal progenitors, affecting their fate

Amandine Di-Cicco[1,2,3], Valérie Petit[1,2], Aurélie Chiche[1,2,3], Laura Bresson[1,2,3], Mathilde Romagnoli[1,2,3], Véronique Orian-Rousseau[4], Maria dM Vivanco[5], Daniel Medina[6], Marisa M Faraldo[1,2,3,7], Marina A Glukhova[1,2,3,7], Marie-Ange Deugnier[1,2,3,7]*

[1]Institut Curie, Paris Sciences et Lettres Research University, Paris, France; [2]CNRS UMR144, Subcellular Structure and Cellular Dynamics - Institut Curie, Paris, France; [3]Equipe Labellisée par La Ligue contre le Cancer, Paris, France; [4]Karlsruhe Institute of Technology, Karlsruhe, Germany; [5]Cell Biology and Stem Cells Unit, CIC bioGUNE, Derio, Spain; [6]Baylor College of Medicine, Houston, United States; [7]Institut national de la santé et de la recherche médicale, Paris, France

**Abstract** HGF/Met signaling has recently been associated with basal-type breast cancers, which are thought to originate from progenitor cells residing in the luminal compartment of the mammary epithelium. We found that ICAM-1 efficiently marks mammary luminal progenitors comprising hormone receptor-positive and receptor-negative cells, presumably ductal and alveolar progenitors. Both cell populations strongly express Met, while HGF is produced by stromal and basal myoepithelial cells. We show that persistent HGF treatment stimulates the clonogenic activity of ICAM1-positive luminal progenitors, controlling their survival and proliferation, and leads to the expression of basal cell characteristics, including stem cell potential. This is accompanied by the induction of *Snai1* and *Snai2*, two major transcription factors triggering epithelial–mesenchymal transition, the repression of the luminal-regulatory genes *Elf5* and *Hey1*, and claudin down-regulation. Our data strongly indicate that paracrine Met signaling can control the function of luminal progenitors and modulate their fate during mammary development and tumorigenesis.

*For correspondence: marie-ange.deugnier@curie.fr

Competing interests: The authors declare that no competing interests exist.

## Introduction

The postnatal development of the mammary gland comprises two distinct morphogenetic events: the growth and branching of epithelial ducts during puberty and the lobulo–alveolar development during pregnancy. The mammary epithelium is embedded in a fatty connective tissue and organized as a bilayer, with a basal layer of myoepithelial cells and a luminal epithelial layer. During lactation, the luminal cells produce milk, whereas myoepithelial cells serve for milk expulsion.

The luminal cell layer is characterized by the expression of keratins 8/18/19 and comprises a subset of hormone-sensing cells that express estrogen, progesterone, and prolactin receptors (ER, PR, and PrlR, respectively) (*Brisken, 2013*). Basal myoepithelial cells express keratins 5 and 14, P-cadherin, the transcription factor p63, and smooth muscle-specific contractile proteins (*Moumen et al., 2011*). This compartment specifically displays Slug/Snail2 (*Guo et al., 2012*; *Nassour et al., 2012*), a key transcription factor coordinating the epithelial–mesenchymal (EMT) transition program (*Thiery et al., 2009*; *Nieto and Cano, 2012*; *Gonzalez and Medici, 2014*).

**eLife digest** Throughout the life of a female mammal, the mammary glands undergo different phases of development to prepare for, and adapt to, feeding offspring. Luminal cells line the inside of branch-like structures throughout the mammary gland and are responsible for producing milk. When the mammary gland grows, new luminal cells develop from a kind of cell called luminal progenitor cells. However, these progenitor cells are also thought to be the source of certain types of breast cancer.

Recently, it has been suggested that luminal progenitor cells display a receptor protein called Met on their surface. When Met and 'co-receptor' proteins bind to a molecule called HGF, this triggers a cascade of signals that can cause certain cells to change their properties. This is known as the epithelial–mesenchymal transition. Although this transition is important for new tissues to develop, it can also result in cancerous tumors forming if it is not correctly controlled. Luminal cells do not produce HGF themselves, which suggests that Met signaling in these cells is triggered by the HGF released from neighboring cells. However, neither the mechanisms behind this signaling nor the effects of signaling on the luminal progenitor cells are well understood.

Di-Cicco et al. set out to identify where Met, its co-receptors and HGF are located in the mouse mammary gland during different phases of development. This revealed that one of the co-receptors—called ICAM-1—can be used as a marker to identify certain types of luminal progenitor cell. Di-Cicco et al. found that these progenitor cells display Met on their surface, and other types of mammary cell—called stromal cells and myoepithelial cells—produce HGF.

When exposed to HGF, luminal progenitor cells grown in culture in the laboratory proliferated and went through the epithelial–mesenchymal transition. These findings suggest that myoepithelial and stromal cells regulate luminal progenitor cells by producing HGF to activate Met signaling in these cells. Such interactions could be of great importance during mammary development and tumorigenesis. The next big challenge will be to determine the circumstances under which luminal progenitor cells stimulated by HGF can give rise to breast cancers. This work will allow us to better define the cell population that should be targeted by anti-cancer drugs.

The adult mammary basal compartment harbors multipotent stem cells able to fully regenerate the gland upon transplantation (*Shackleton et al., 2006*; *Stingl et al., 2006*). The luminal cell layer contains clonogenic cells, the luminal progenitors (*Asselin-Labat et al., 2007*; *Sleeman et al., 2007*). This population is heterogeneous, consisting of hormone receptor-positive and hormone receptor-negative clonogenic cells that are considered to be functionally distinct ductal and alveolar progenitors (*Beleut et al., 2010*; *Regan et al., 2012*; *Shehata et al., 2012*; *Fu et al., 2014*). Considerable interest has recently focused on luminal progenitors. According to lineage-tracing studies, these cells are able to drive the expansion of the luminal cell population in mammary ducts and alveoli during postnatal development (*Van Keymeulen et al., 2011*; *Fu et al., 2014*; *Rios et al., 2014*). Moreover, luminal progenitors are thought to be at the origin of the triple-negative, basal-like, Brca1-associated breast cancers, indicating that they display phenotypic plasticity and an ability to upregulate basal markers (*Lim et al., 2009*; *Molyneux et al., 2010*; *Proia et al., 2011*). Of note, ectopic expression of Snail2 endows luminal progenitors with basal cell features (*Guo et al., 2012*), supporting an important role of EMT in the regulation of luminal progenitor plasticity.

The Met tyrosine kinase receptor and its major ligand, HGF, are known to control numerous epithelial cell functions and trigger EMT (*Trusolino et al., 2010*; *Gherardi et al., 2012*). Met signaling has been implicated in mammary development and tumorigenesis. Although not investigated in detail yet, Met-deficient mammary glands of adult MMTV-Cre:Met[f/f] mice showed defects in branching morphogenesis (*Garner et al., 2011*). Aberrant Met activation occurs in breast cancers, in particular in the triple-negative, basal-like subtype (*Gastaldi et al., 2010*; *Ponzo and Park, 2010*). Expression of activated Met or over-expression of HGF in mouse mammary gland induce tumors, including those of the basal-like subtype (*Graveel et al., 2009*; *Ponzo et al., 2009*; *Holland et al., 2013*; *Knight et al., 2013*). Several in vitro studies performed with mammary organoids or immortalized cell lines have implicated HGF/Met signaling in the regulation of mammary epithelial cell growth, morphogenesis, and differentiation (*Soriano et al., 1995*; *Yang et al., 1995*; *Haslam et al., 2008*; *Lee et al., 2011*).

One recent work indicated that luminal progenitors express Met and that, when forced to express HGF, display an enhanced in vivo regenerative potential, a property attributed to basal-type stem cells (*Gastaldi et al., 2013*). However, in situ, mammary luminal cells do not express HGF, suggesting that Met signaling in mammary epithelial cells is activated in a paracrine manner. HGF-mediated paracrine and autocrine Met stimulation can trigger distinct biological responses in epithelial cells (*Mai et al., 2014*). How Met-expressing luminal progenitors respond to the physiological paracrine action of HGF has not yet been investigated at the cellular and molecular levels.

Met activation requires adhesion molecules as co-receptors. Two co-receptors for Met have been identified in hepatocytes: CD44v6, the CD44 isoform containing the exon v6 and ICAM-1 (CD54), a member of the immunoglobulin superfamily of cell adhesion molecules (*Orian-Rousseau et al., 2002*; *Olaku et al., 2011*). In this study, we investigated the expression of ICAM-1, CD44v6, HGF, and Met in the mouse mammary epithelium. We found that ICAM-1 is a useful marker for obtaining highly enriched preparations of clonogenic luminal progenitors that express *Met*, and that HGF is produced by both stromal and myoepithelial cells. Using ICAM-1 as a surface marker, we further characterized Met-expressing luminal progenitors and analyzed their cellular and molecular responses to HGF. Our data suggest that HGF can act as a paracrine regulator of luminal progenitor function and fate during mammary development and tumorigenesis.

## Results

### ICAM-1 expression discriminates the basal/myoepithelial cell compartment and two distinct populations of luminal cells

We examined ICAM-1 expression at representative stages of mammary gland development, including puberty, maturity, early, and late gestation. Freshly isolated mammary cells, double-stained for CD24 and ICAM-1, were analyzed by flow cytometry (*Figure 1A*, *Figure 1—figure supplement 1A*). From puberty to late gestation, ICAM-1 was differentially expressed in the mammary epithelium. Staining for this marker identified three epithelial populations, negative (ICAM1-neg) or displaying low (ICAM1-low) or high (ICAM1-hi) levels of expression (*Figure 1A*, *Figure 1—figure supplement 1B*). Gene expression analysis by qPCR showed that, at all stages of development, K5-expressing basal myoepithelial cells were located in the ICAM1-hi population, while K8-positive luminal cells were distributed in ICAM1-neg and ICAM1-low-cell subsets (*Figure 1B*). Thus, ICAM-1 can be used to identify the basal cell compartment, and separate two populations of luminal cells, referred to hereafter as Lu-pos and Lu-neg cells.

Throughout development, the basal myoepithelial cell population constitutively expressed ICAM-1 at high level (*Figure 1A*). By contrast, the luminal cells displayed temporal changes in ICAM-1 expression (*Figure 1A*, *Figure 1—figure supplement 1C*). In virgin mice, most of the luminal cells were negative for ICAM-1 at puberty, whereas 25–30% of the luminal cells expressed ICAM-1 in the mature gland of 11 to 14-week-old cycling females. Notably, the Lu-pos cell subset was amplified early in gestation (P-8d), during the expansion of the alveolar buds. Later, at P-16d, when the alveoli were well developed, almost the entire luminal compartment was positive for ICAM-1. Thus, in the luminal cell population, ICAM-1 expression is developmentally regulated and correlates temporally with changes in hormone signaling.

### ICAM-1 expression delineates a luminal cell population highly enriched in clonogenic progenitors

We investigated the functional properties of the luminal cell populations separated on the basis of ICAM-1 expression, by plating purified Lu-pos and Lu-neg cells at low density and assessing their ability to form colonies in two-dimensional cultures. In adult virgin mice, almost all the colony-forming activity was associated with the Lu-pos cell subset, demonstrating that ICAM-1 expression delineates a luminal population highly enriched in clonogenic progenitors (*Figure 1C*). Notably, ICAM-1 efficiently marked mammary luminal progenitors from adult virgin Balb/c (*Figure 1C*), C57Bl/6J (*Figure 1—figure supplement 2A*) and Blg-Cre; R26 mice (*Figure 1—figure supplement 2B–E*). As expected from previous findings (*Selbert et al., 1998*; *Molyneux et al., 2010*), LacZ activity, as assessed by X-gal staining, was undetectable in basal myoepithelial cells and present in a subset of luminal cells in the mammary gland of adult virgin Blg-Cre; R26 females (*Figure 1—figure supplement 2C*). An analysis of Cre expression by qPCR indicated that *Blg* (β-Lactoglobulin) drove recombination

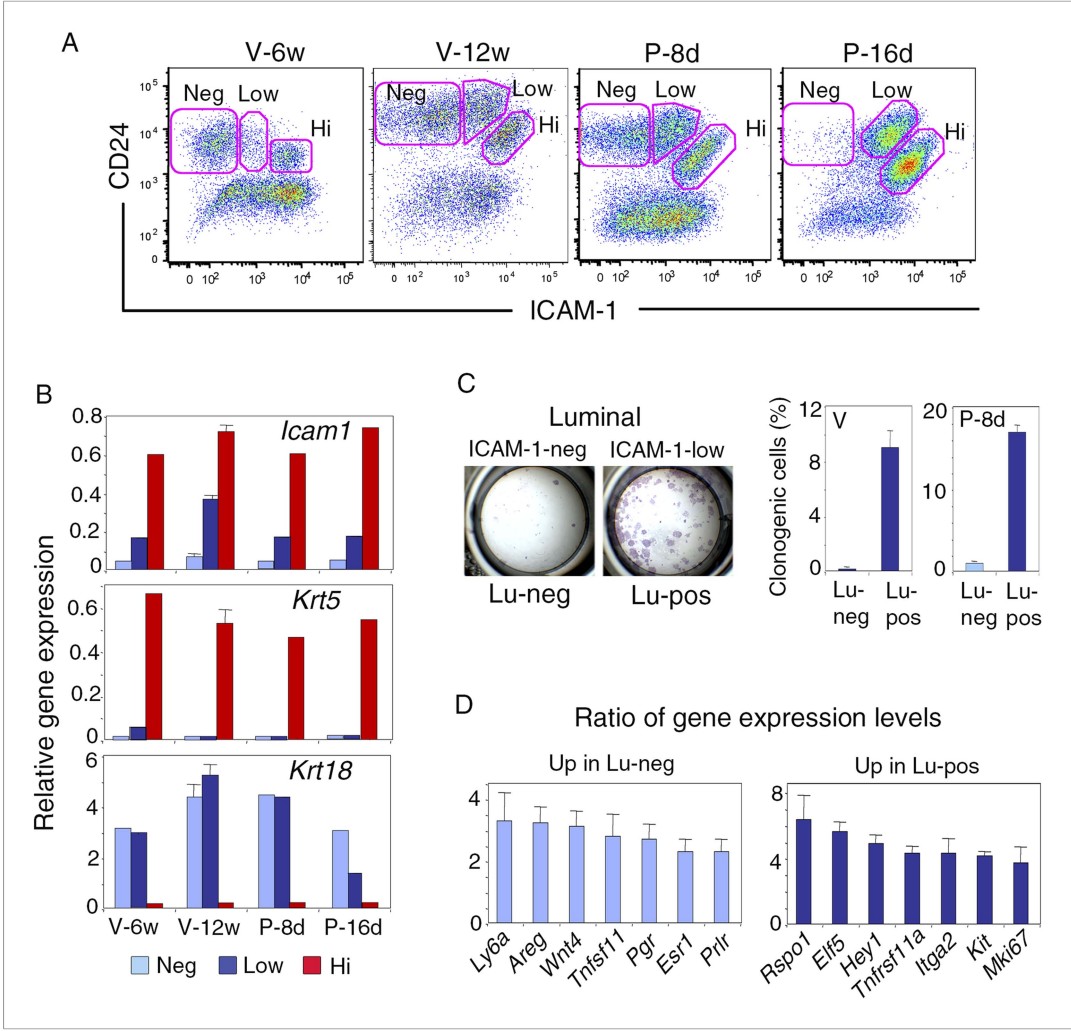

**Figure 1**. ICAM-1 expression discriminates basal and luminal cell compartments and defines a luminal population highly enriched in clonogenic progenitors. (**A**) Flow cytometry dot plots showing CD24 and ICAM-1 expression in cells isolated from mouse mammary glands taken at representative stages of development: V-6w, 6-week-old virgin; V-12w, 12-week-old virgin; P-8d, 8-day pregnant mice; P-16d, 16-day pregnant mice. Within the CD24-positive epithelial cell population, ICAM-1 discriminated three distinct fractions, negative (Neg), low-expressing (Low), and high-expressing (Hi) cells. (**B**) Levels of *Icam1* and lineage-specific gene expression in ICAM1-neg, ICAM1-low, and ICAM1-hi epithelial cells as determined by q-PCR analysis. Cells were isolated from mammary glands at different stages of development, as shown in panel **A**. The values were normalized to *Gapdh* expression and represent mean values from at least two distinct cell preparations. Data obtained with adult virgin mice (V-12w) are from four independent groups of cell samples and presented as mean ±S.E.M. (**C**) Colony formation by ICAM1-neg (Lu-neg) and ICAM1-low (Lu-pos) mammary luminal cells. Left panel: hematoxylin and eosin (H&E) staining of clonal colonies after 8 days in culture. Right panel: percentages of clonogenic cells. Cells were isolated from mature virgin mice (V) and early pregnant females (P-8d). The results are from two (P-8d) or three (V) independent cell preparations (each of which with three separate wells), and presented as mean values ±S.E.M. (**D**) q-PCR analysis of relative gene expression levels in Lu-neg and Lu-pos cells isolated from mammary glands of mature virgin females. Mean ratios (±S.E.M) of values normalized to *Gapdh* expression are shown. Lu-neg/Lu-pos and Lu-pos/Lu-neg ratios are presented in left and right panels, respectively. Results are from three independent cell preparations.

The following source data and figure supplements are available for figure 1:

**Source data 1**.

**Figure supplement 1**. Gating procedure for flow cytometry analysis.

*Figure 1. continued on next page*

*Figure 1. Continued*

**Figure supplement 2**. Isolation of mammary luminal progenitors from adult virgin C57Bl/6J and Blg-Cre; R26 females using ICAM-1.

primarily in the luminal progenitor population identified by ICAM-1 (*Figure 1—figure supplement 2D*). Accordingly, 98% of the colonies formed by ICAM1-expressing Blg-Cre; R26 luminal cells were LacZ-positive (*Figure 1—figure supplement 2E*).

We next compared the molecular characteristics of the luminal cell populations defined by ICAM-1 by analyzing the expression of a panel of genes by qPCR (*Figure 1D*, *Figure 1—figure supplement 2F*). The non-clonogenic Lu-neg population had high level of transcripts for the hormone receptors, ERα, PR, and PrlR, and for genes encoding secreted hormonal mediators implicated in the control of mammary development such as amphiregulin (encoded by *Areg*), RankL (encoded by *Tnfsf11*) and Wnt4. These genes, along with *Ly6a* (encoding Sca-1), are characteristic of mature mammary luminal cells (*Mulac-Jericevic et al., 2003*; *Kendrick et al., 2008*; *Cai et al., 2014*). The clonogenic Lu-pos population expressed *Mki67*, a marker of cell proliferation, more strongly than the Lu-neg fraction. It exhibited high levels of expression for genes encoding essential regulators of mammary development including Elf5, the Notch effector, Hey1, RANK (encoded by *Tnfrsf11a*), and the local mediator R-spondin1 (encoded by *Rspo1*) (*Fata et al., 2000*; *Bouras et al., 2008*; *Oakes et al., 2008*; *Cai et al., 2014*). Lu-pos cells also over-expressed *Kit* and *Itga2*, two markers previously used to obtain mammary cell populations enriched in luminal progenitors (*Regan et al., 2012*; *Shehata et al., 2012*).

The Lu-neg and Lu-pos cell populations isolated from the mammary glands of females in early pregnancy had functional properties and gene expression patterns similar to those purified from adult virgin glands (*Figure 1D*, *Figure 2—figure supplement 1A*). ICAM-1 thus appeared to be a robust surface marker for the enrichment in mammary luminal progenitors from adult virgin and early pregnant females.

## The luminal progenitors identified by ICAM-1 express Met whereas myoepithelial and stromal cells produce HGF

We then used q-PCR to examine the expression of Met and its major cytoplasmic effector, Gab1, in the mammary cell populations separated by ICAM-1. In adult virgin mice, Met transcript levels were five and ten times higher in the Lu-pos population than in the Lu-neg and myoepithelial cells, respectively; *Gab1*, like *Met*, was strongly expressed only in the Lu-pos fraction (*Figure 2A*). Similarly, the clonogenic Lu-pos population isolated from the mammary glands of early pregnant females was greatly enriched in Met-expressing cells (*Figure 3—figure supplement 1B*). Thus, in adult virgin females and during early pregnancy, the target cells for Met signaling mostly belonged to the luminal progenitor-enriched population defined by ICAM-1 expression.

As expected (*Niranjan et al., 1995*; *Yang et al., 1995*), HGF, the major Met ligand, was strongly expressed in the CD24-negative Pdgfr-expressing stromal cell population (*Figure 2B*). In addition, HGF was expressed at high level in the mammary basal cell population of mature virgin and early pregnant mice (*Figure 2A,B*, *Figure 2—figure supplement 1B*), suggesting paracrine interactions between basal myoepithelial cells and luminal progenitors.

As both ICAM-1 and CD44v6 may serve as co-receptors for Met (*Olaku et al., 2011*), we analyzed the expression of CD44v6 in freshly isolated Lu-pos and Lu-neg cells. Most cells of both populations were stained with the anti-CD44v6 antibody (*Figure 2—figure supplement 1C*). Thus, CD44v6 is broadly expressed in the luminal compartment and cannot, therefore, be used for the enrichment in Met-expressing progenitors.

## HGF stimulates the clonogenic activity of luminal progenitors and promotes acquisition of basal-specific traits

We used ICAM-1 to isolate Met-expressing luminal progenitors and investigated their response to the paracrine action of HGF. Purified Lu-pos cells were cultured in suspension in the presence of 2% Matrigel as previously described (*Spike et al., 2012*; *Chiche et al., 2013*). We found that HGF-treated

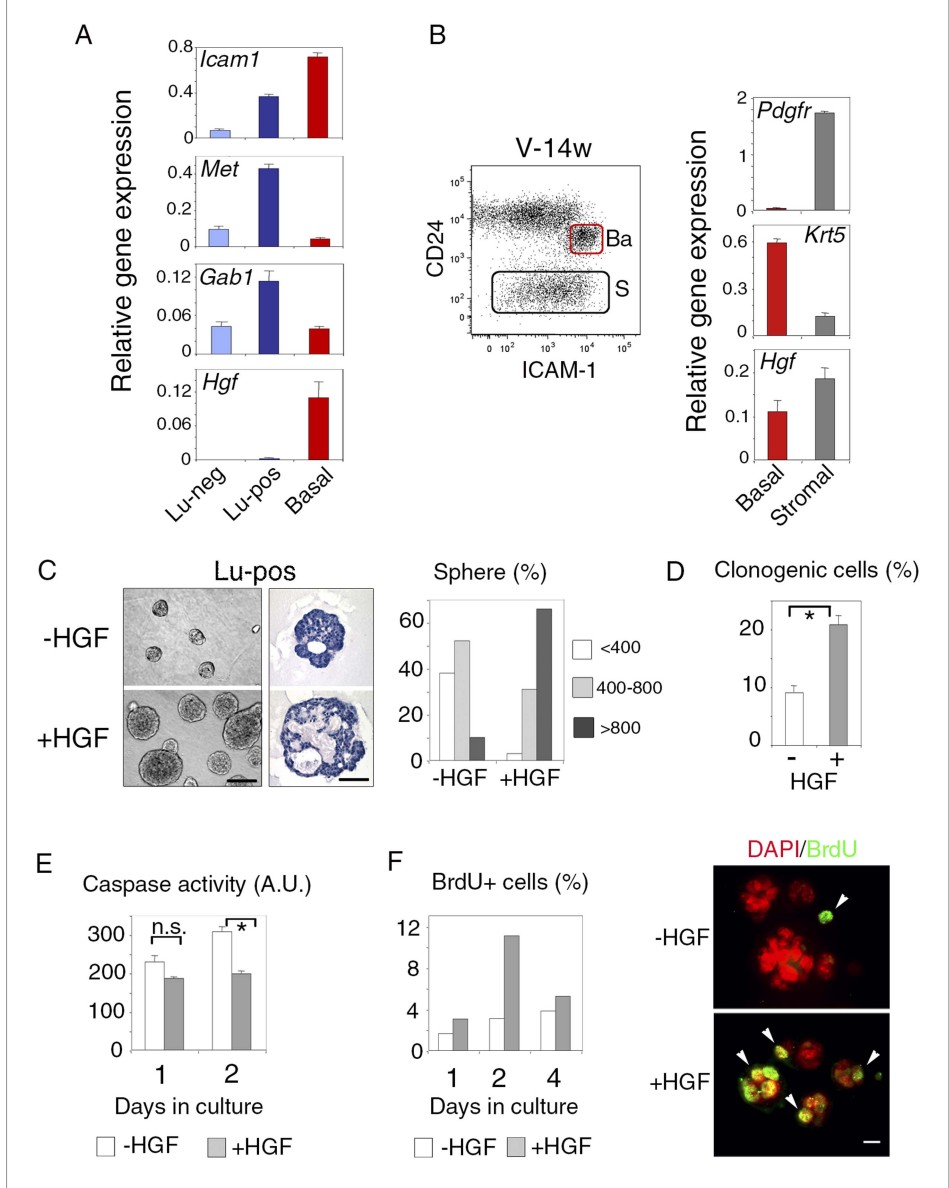

**Figure 2**. ICAM-1 identifies Met-expressing clonogenic luminal progenitors. (**A**) *Icam-1*, *Met*, *Gab1*, and *Hgf* expression in Lu-neg, Lu-pos, and basal/myoepithelial cells isolated from mammary glands of mature virgin mice by flow cytometry. The q-PCR values were normalized to *Gapdh* expression and represent mean values ±S.E.M from three independent preparations. (**B**) Levels of *Hgf* expression in stromal and basal/myoepithelial cells. Left panel: flow cytometry dot plot showing mammary basal (Ba) and stromal (S) cell compartments isolated from a 14-week-old virgin mice. Right panel: q-PCR analysis of *Hgf* expression in basal and stromal cells. The values were normalized to *Gapdh* expression and represent mean values ±S.E.M from three independent preparations. (**C**) Characteristics of HGF-treated and untreated spheres derived from purified Lu-pos cells cultured in the absence or presence of HGF for 10 days. Left panels: Representative phase contrast images of HGF-treated and untreated spheres (bar, 400 μm) and H&E staining of sections through HGF-treated and untreated spheres (bar, 150 μm). Right panel: Sphere size distribution (in arbitrary units) in HGF-treated and untreated cultures. At least 250 spheres were analyzed per conditions. (**D**) Average percentages (±S.E.M) of clonogenic cells in non-stimulated and HGF-stimulated cultures of Lu-pos cells. Data from two independent cell preparations (each of which with three separate wells analyzed at day 10) are shown. *p < 0.0004. (**E**) Caspase-3 and caspase-7 activity in non-stimulated and HGF-stimulated Lu-pos cells after 1 and 2 days in culture as measured by a luminescent assay. Data are the mean (±S.E.M) of three measurements from separate wells. *p < 0.003 at day 2, not significant (n.s) at day 1. A.U., arbitrary unit. (**F**) BrdU incorporation in Lu-pos cells grown in the absence or presence of HGF for 1, 2, and 4 days. Left panel: percentage of BrdU-positive cells. Mean values from two distinct cell preparations are shown. Right panel: representative images of cells

*Figure 2. continued on next page*

*Figure 2. Continued*
cytocentrifuged and immunostained with anti-BrdU antibody after 2 days in culture. Nuclei were counterstained with DAPI. Arrowheads indicate proliferating BrdU-positive cells. Bar, 20 μm.
The following source data and figure supplement are available for figure 2:

**Source data 1**.
**Figure supplement 1**. Molecular and phenotypic characteristics of luminal progenitors isolated from virgin and early pregnant females.

Lu-pos cells formed enlarged spheres after 10 days in culture (*Figure 2C*). These spheres had multiple lumens, with an overall lumen space larger than that of the untreated spheres (*Figure 2C*).

Quantitative analysis revealed that HGF-treated cultures of purified Lu-pos cells contained twice as many spheres as non-stimulated cultures (*Figure 2D*), suggesting a role for HGF in favoring cell survival and/or proliferation. Measurements of caspase activity showed that HGF-treated cells had lower levels of activated caspase-3 and caspase-7 than untreated cells after 2 days in culture (*Figure 2E*). BrdU-incorporation assays performed after 1, 2, and 4 days in culture indicated that HGF treatment increased cell proliferation, particularly on day 2 (*Figure 2F*). Thus, in 3D cultures, HGF stimulated the clonogenic activity of ICAM1-expressing luminal progenitors, by promoting cell survival and proliferation early in culture.

We then investigated the long-term effects of HGF on purified luminal progenitors. Using q-PCR, we analyzed proliferation and lineage-specific gene expression in Lu-pos cells cultured in the presence or absence of HGF for 9–10 days (*Figure 3A*). *Mki67* was down-regulated in HGF-stimulated cultures whereas *Cdkn1a*, which encodes p21, a negative regulator of the cell cycle (*Abbas and Dutta, 2009*), was upregulated by a factor of 2.5. The basal-specific keratin, Krt5, undetectable in untreated spheres, was clearly expressed in HGF-stimulated spheres, whereas no significant decrease was observed in expression of the luminal-specific keratin, Krt18. Thus, in the long-term, HGF/Met signaling biased luminal progenitors toward a basal cell fate while attenuating cell cycle progression.

Consistent with the q-PCR data, immunofluorescence labeling showed that the percentages of K5-positive cells in spheres stimulated with HGF for 9–10 days were six times higher that that of untreated cultures (*Figure 3B*). Numerous K5-expressing cells stained positive for K8 (*Figure 3B*), and none contained the myoepithelial-specific protein α-SMA (*Figure 3—figure supplement 1*). Interestingly, BrdU incorporation assays showed that, unlike K5-negative luminal cells, K5-expressing cells did not cycle following HGF stimulation (*Figure 3C*).

Thus, paracrine Met activation controls the survival and proliferation of ICAM1-expressing luminal progenitors and favors the acquisition of basal-specific keratin.

## Luminal progenitors activate EMT program and repress luminal-specific regulatory genes upon stimulation by HGF

To prove that K5-expressing cells in HGF-treated spheres originated from luminal progenitors, we isolated Lu-pos cells from Blg-Cre; R26 mouse mammary glands and stimulated them with HGF for 9–10 days. As in Balb/c mice, Met was strongly expressed in the Blg-Cre; R26 luminal progenitor population purified with ICAM-1 (*Figure 4—figure supplement 1A*). Notably, in HGF-stimulated cultures, 85% of the K5-positive cells expressed LacZ and were therefore of luminal origin (*Figure 4A*, *Figure 4—figure supplement 1B*). Control HGF-treated basal cells did not stain positive for X-gal.

We investigated the molecular events induced by the paracrine action of HGF in luminal progenitors, by analyzing the expression of basal-specific and luminal-specific regulatory genes in untreated and HGF-treated spheres derived from purified Lu-pos cells (*Figure 4B*). Upon stimulation with HGF, the expression of essential regulators of the luminal cell fate such as *Elf5*, *Hey1*, and *Gata3* was repressed, whereas that of several basal-specific genes increased markedly. These genes included *Cdh3* (encoding P-cadherin), *Trp63*, and the EMT inducer *Snai2*. Other genes associated with EMT program—*Snai1*, *Sox9*, and *Cdh2* (encoding N-cadherin)—were also upregulated. In agreement with gene expression data, immunodetection studies revealed that HGF-treated spheres contained

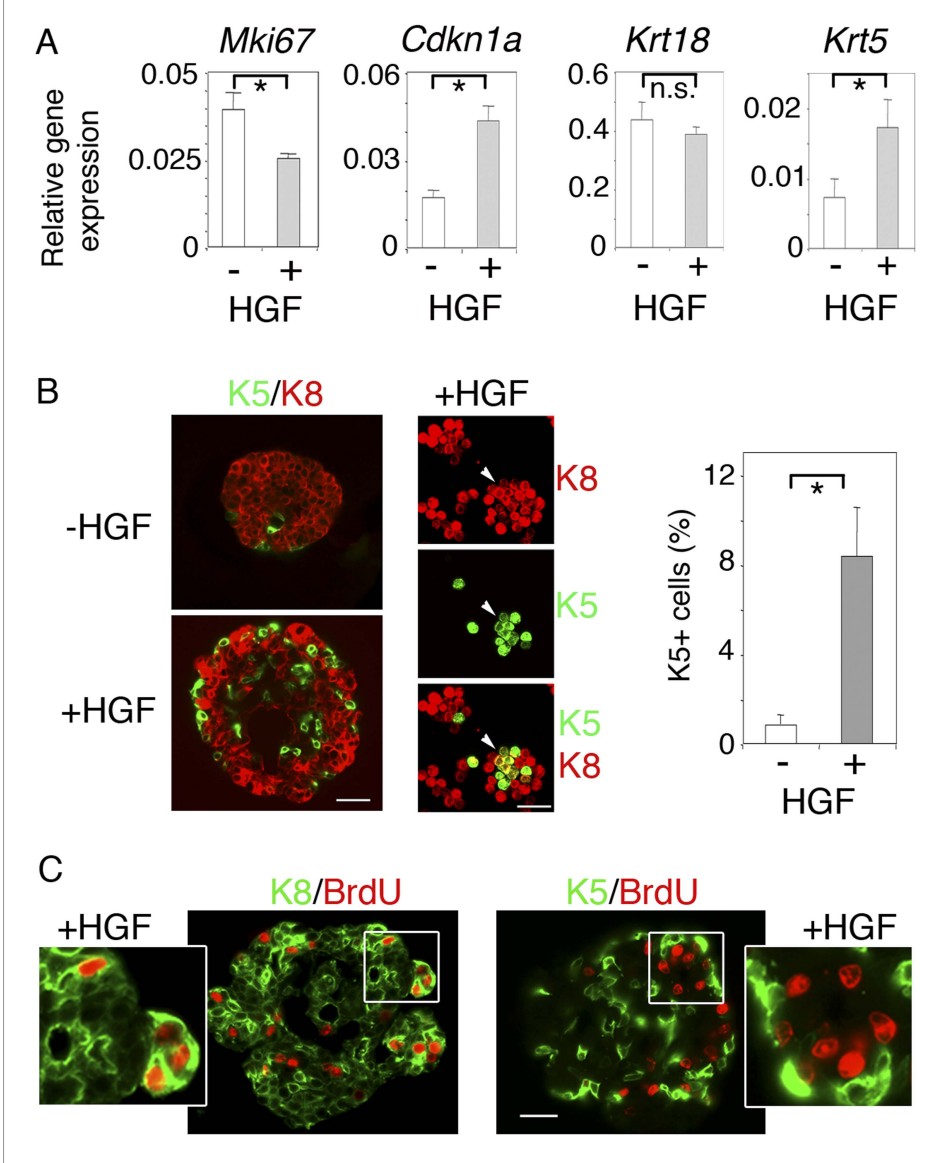

**Figure 3**. HGF promotes acquisition of basal-specific keratins in luminal progenitors identified by ICAM-1. (**A**) Analysis of *Mki67, Cdkn1, Krt18* and *Krt5* expression in spheres derived from untreated and HGF-treated Lu-pos cells, as determined by q-PCR. Lu-pos cells were isolated from mammary glands of mature virgin mice. The values, normalized to *Gapdh* expression, represent mean values ±S.E.M. from at least three independent sphere preparations harvested after 10 days in culture. *p < 0.05, p < 0.001, p < 0.007 for *Mki67, Cdkn1,* and *Krt5,* respectively. n.s, not significant (*Krt18*). (**B**) Double immunofluorescence labeling of non-stimulated and HGF-stimulated Lu-pos cells grown for 10 days. Left panel: K8 and K5 staining of spheres sections. Bar, 60 µm. Middle panel: K8 and K5 staining of cytocentrifuged cells derived from HGF-stimulated cultures. Arrowheads indicate a group of double-positive K5/K8 cells. Bar, 45 µm. Right panel: Average percentages (±S.E.M) of K5-expressing cells in 10- to 12-day-old spheres derived from untreated and HGF-treated Lu-pos cells. 1000 cells at least were counted per sample. Percentages of K5-positive cells in untreated and HGF-treated cultures were 0.8% ± 0.4 and 8.4% ± 2.2, respectively. Data are from four independent cell preparations. *p < 0.002. (**C**) BrdU incorporation in 10-day-old HGF-treated spheres. Sections were stained with anti-BrdU antibody combined either to anti-K8 (left panels) or anti-K5 (right panels) antibodies. Pictures at low magnification and enlarged views of defined areas are shown. Bar, 45 µm.

The following source data and figure supplement are available for figure 3:

**Source data 1**.

**Figure supplement 1**. α-SMA expression in untreated and HGF-treated spheres.

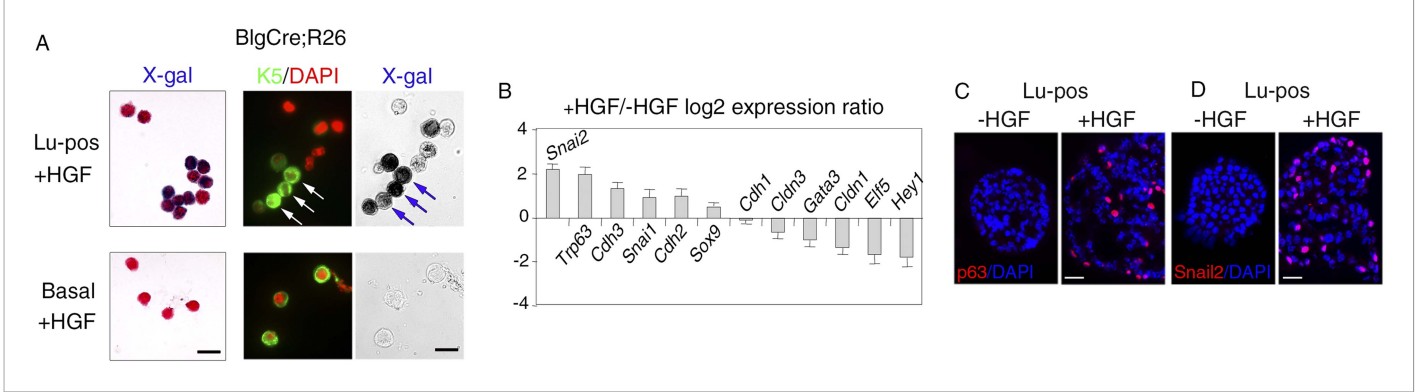

**Figure 4**. HGF triggers activation of EMT program in ICAM1-expressing luminal progenitors. (**A**) K5 and β-galactosidase expression in cells isolated from HGF-treated spheres. Lu-pos (upper panels) and basal cells (lower panels) were purified from mammary glands of mature virgin Blg-Cre; R26 mice and stimulated with HGF for 13 days. Left panels: X-gal staining with fast red counterstaining. Bar, 15 µm. Middle and right panels: correlated images of K5 immunostaining and X-gal staining. Arrows indicate LacZ-positive cells expressing K5. Quantitative data are shown in *Figure 4—figure supplement 1B*. Bar, 10 µm. (**B**) Comparative expression levels of basal-specific, epithelial–mesenchymal transition (EMT)-associated and luminal-specific genes in spheres derived from untreated and HGF-treated Lu-pos cells as determined by q-PCR. Lu-pos cells isolated from mammary glands of mature virgin Balb/c mice were cultured in the absence or presence of HGF for 10 days. Results are expressed as Log$_2$ ratios of values normalized to *Gapdh* and represent mean values ±S.E.M. from at least three independent sphere preparations. The comparator values were those obtained with untreated spheres. (**C-D**) Immunodetection of p63 (**C**) and Snail2 (**D**) in sections through untreated and HGF-stimulated spheres. Nuclei were counterstained with DAPI. Bar, 25 µm.

The following source data and figure supplement are available for figure 4:

**Source data 1**.

**Figure supplement 1**. Met expression in luminal progenitors isolated from Blg-Cre; R26 mice.

numerous p63- and Snail2-positive cells, whereas untreated cells were negative for these basal-specific markers (*Figure 4C,D*).

*Snai2* and *Snai1* can directly repress the transcription of *Cdh1* (encoding E-cadherin) and claudins (reviewed in *Thiery et al., 2009*; *Gonzalez and Medici, 2014*). Noticeably, *Cdh1* expression was not significantly reduced upon HGF treatment, whereas claudin-1 and claudin-3 transcript levels decreased strongly (*Figure 4B*). Consistent with the q-PCR data, immunofluorescence staining revealed E-cadherin in the vast majority of cell–cell contacts in untreated and HGF-treated spheres (*Figure 5A*). Notably, numerous Snail2-positive cells present in the HGF-stimulated spheres displayed E-cadherin at their surface (*Figure 5B*). Claudin-1 distribution was more heterogeneous than that of E-cadherin, but it was clearly altered following HGF treatment. Untreated spheres contained a majority of cells with claudin-1 at their junctions, whereas HGF-treated spheres displayed large cell areas lacking claudin-1 expression (*Figure 5C*).

Thus, the persistent stimulation of Met signaling by exogenous HGF in luminal progenitors repressed luminal-regulatory gene expression, upregulated basal-specific markers, and triggered an EMT program, including *Snai2* and *Snai1* expression and decrease in claudin levels.

## Persistent stimulation by HGF is required to sustain effects on luminal progenitors

To examine whether the HGF/Met signaling effects on luminal progenitors persist after HGF withdrawal, primary HGF-treated spheres obtained from purified Lu-pos cells were dissociated and then cultured either with or without HGF for 10 additional days. Both clonogenic activity and level of expression of the basal-specific keratin Krt5 were much lower in secondary mammospheres deprived of HGF (*Figure 6A,B*). Consistently, immunofluorescence stainings revealed that, like untreated control cultures, secondary mammospheres deprived of HGF contained less than 1% K5-positive cells (*Figure 6C,D*). HGF-treated secondary mammospheres contained twice more K5-positive cells than HGF-treated primary spheres. Many K5-positive cells co-expressed K8, as in primary spheres,

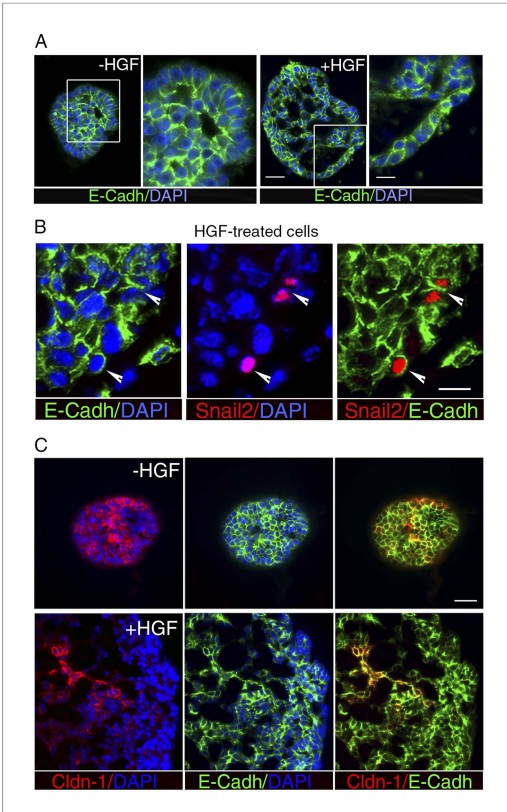

**Figure 5.** HGF treatment perturbs cell–cell adhesion in ICAM1-expressing luminal progenitors. (**A**) Immunofluorescence labeling of sphere sections with anti-E-cadherin antibody. Low- and high-magnification views of untreated (left panels) and HGF-treated (right panels) spheres. Nuclei were stained with DAPI. Bars, 60 μm and 30 μm. (**B**) Double immunodetection of Snail2 and E-cadherin in a HGF-treated sphere. Arrowheads indicate Snail2-positive cells expressing E-cadherin at their surface. Nuclei were stained with DAPI. Bar, 20 μm. (**C**) Double immunofluorescence labeling of sphere sections with anti-claudin-1 and anti-E-cadherin antibodies. Nuclei were stained with DAPI. Bar, 75 μm.

(*Figure 6C*). Experiments performed with mammary luminal progenitors purified from BlgCre; R26 mice confirmed that, in HGF-treated secondary mammospheres, K5-expressing cells were derived from LacZ-positive luminal cells (*Figure 6E*). Gene expression analysis showed that *Trp63*, *Snai2*, and *Cdkn1a* were upregulated whereas *Hey1*, *Elf5*, and *Mki67* were downregulated only in HGF-treated secondary mammospheres (*Figure 6F*).

Thus, the continuous activation of HGF/Met signaling is required to sustain the response of luminal progenitors in vitro and leads to the accumulation of cells with basal characteristics.

## HGF stimulation confers regenerative potential on luminal progenitors

To compare the stem cell activity of untreated and HGF-treated luminal progenitors, we transplanted either single-cell suspensions or intact spheres into cleared mammary fat pads of recipient mice (*Figure 7*). Untreated cells isolated from primary spheres failed to repopulate the fat pad or gave rise to very limited outgrowths. In contrast, HGF-treated cultures formed ductal-type outgrowths in seven of the eight transplanted fat pads (*Figure 7A*, *Supplementary file 1*). Serial passages of luminal progenitor-derived mammospheres stimulated by HGF led to an important enrichment in K5-positive cells (*Figure 6C*); we therefore compared the regenerative potential of secondary spheres from untreated and HGF-stimulated cultures. Strikingly, these transplantation assays showed that only HGF-treated spheres were able to form outgrowths (*Figure 7B*). 6-week-old outgrowths were composed of well-organized ducts and growing buds comprising K8-positive luminal cells and basally located α-SMA-expressing myoepithelial cells (*Figure 7C,D*).

These data demonstrate that paracrine Met activation induced multipotent stem cell properties in luminal progenitors.

## Met-expressing luminal progenitors comprise hormone receptor-positive and hormone receptor-negative cells

Differential Sca-1 expression in the luminal compartment has been reported to delineate populations enriched in hormone receptor-positive and hormone receptor-negative cells, both containing clonogenic progenitors (*Sleeman et al., 2007*; *Regan et al., 2012*). We characterized Met-expressing luminal progenitors further, by analyzing ICAM-1 and Sca-1 distribution in the luminal compartment at different developmental stages by flow cytometry (*Figure 8A*, *Figure 8—figure supplement 1A*).

In adult virgin mammary gland, ICAM-1 separated both Sca-1-positive and Sca-1-negative cell populations, revealing four cell subsets, Lu1 (ICAM1-neg/Sca-1-pos), Lu2 (ICAM1-pos/Sca-1-pos), Lu3 (ICAM1-neg/Sca-1-neg), and Lu4 (ICAM1-pos/Sca-1-neg) accounting for 39.8 ± 4.6%, 9.1 ± 1.4%, 15.0 ± 2.5%, and 25.4 ± 6.1% of the luminal compartment, respectively (*Figure 8A*). We assessed the

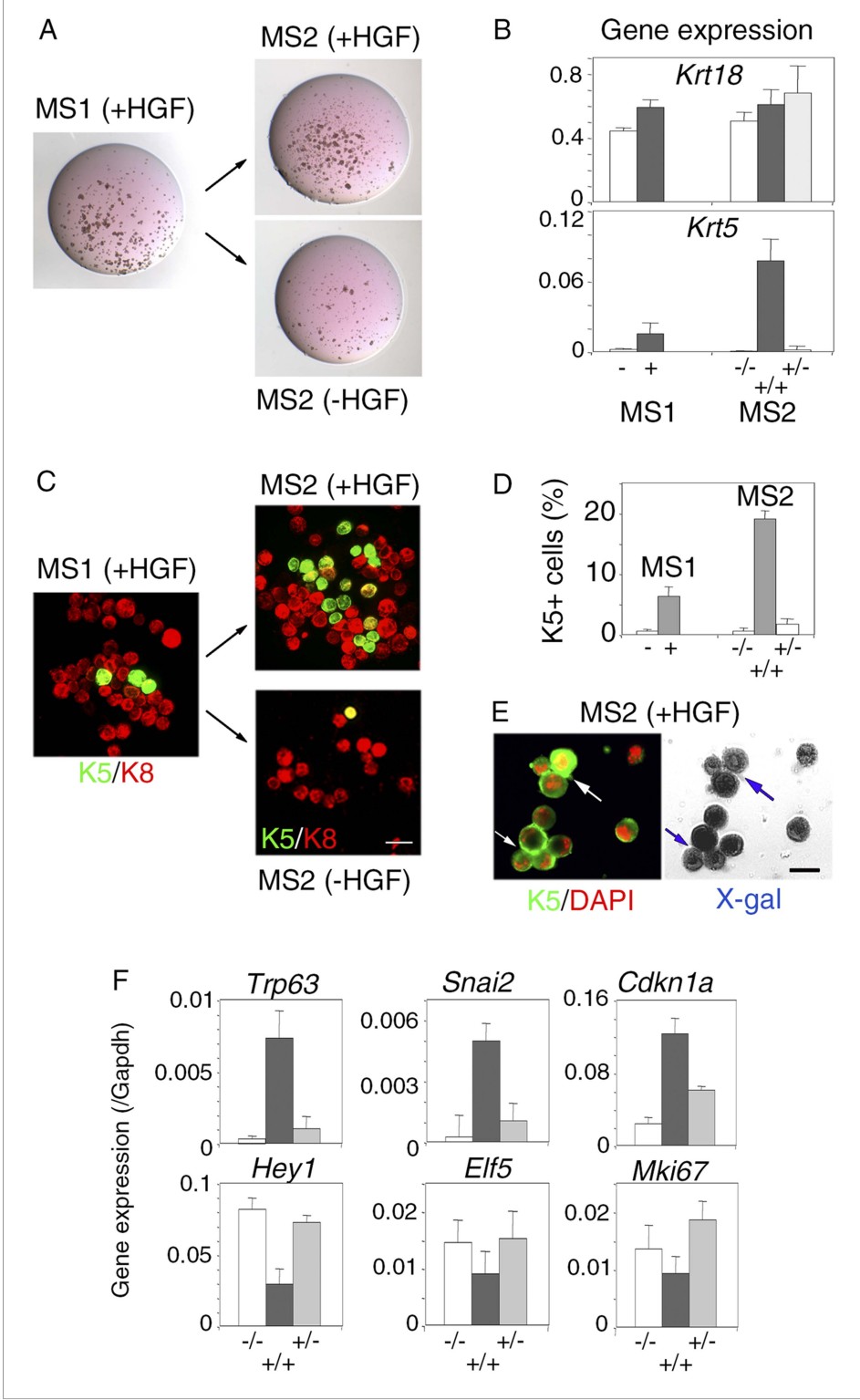

**Figure 6**. Persistent stimulation with HGF is required to sustain effects on luminal progenitors. (**A**) Microphotographs of primary (MS1) and secondary (MS2) mammospheres derived from purified Lu-pos cells. Primary spheres obtained after 11 days of culture in the presence of HGF were dissociated and 5000 cells were replated and grown either in the presence or absence of HGF for an additional period of 10 days. (**B**) *Krt18* and *Krt5* expression levels in MS1 and MS2 cultures derived from Lu-pos cells as determined by q-PCR. HGF-treated MS2 cultures (+) grown either in the presence or absence of HGF are labeled +/+ and +/−, respectively. Untreated primary (−) and secondary (−/−) spheres of Lu-pos cells served as controls. The values were normalized to *Gapdh* expression. Data are presented as mean values ±S.E.M. of three
*Figure 6. continued on next page*

*Figure 6. Continued*

independent experiments. (**C**) K5-expressing cells in MS1 and MS2 cultures derived from Lu-pos cells. Immunodetection of K5 and K8 in cytocentrifuged cells isolated from MS1 and MS2. Bar, 30 μm. (**D**) Percentages of K5-expressing cells in MS1 and MS2 cultures according to their treatment with HGF. 2000 cells were counted per analyzed cytospots. Data from two independent experiments (three separate cytospots) are presented as mean values ±S.E.M. (**E**) Correlated images of K5 immunostaining and X-gal staining in cells isolated from HGF-treated secondary spheres. Lu-pos cells were purified from mammary glands of mature virgin Blg-Cre; R26 mice and continuously stimulated with HGF. Arrows indicate two groups of LacZ-positive cells expressing K5. Bar, 15 μm. (**F**) Modulation of *Trp63*, *Snai2*, *Hey1*, *Elf5*, *Cdkn1a*, and *Mki67* expression levels in MS2 cultures derived from Lu-pos cells as determined by q-PCR. The q-PCR values were normalized to *Gapdh* expression. Results are shown as mean values ±S.E.M. of three independent experiments.
The following source data is available for figure 6:
**Source data 1**.

---

colony-forming potential of these cell subsets. Lu1 was considered as non-clonogenic (<0.5% colony-forming cells), Lu3 cells, negative for ICAM-1, were poorly clonogenic whereas both Lu2 and Lu4 cells, positive for ICAM-1, were highly clonogenic (*Figure 8B*).

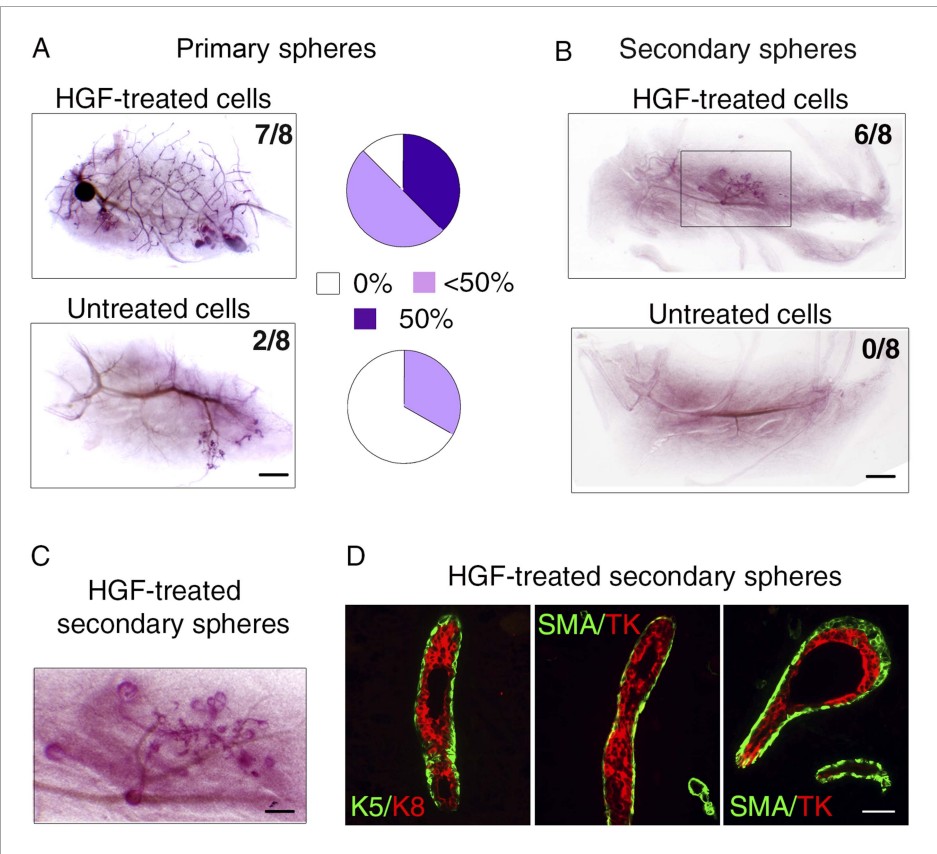

**Figure 7**. HGF-treated luminal progenitors display regenerative potential. (**A**) Regenerative properties of cells isolated from untreated and HGF-treated cultures of luminal progenitors. Primary spheres were dissociated and single cell suspensions were transplanted at 2000, 1000, and 500 cells/fat pad. Left panels: Representative images of carmine-stained outgrowths obtained 12 weeks after transplantation of 2000 cells and take rates. Bar, 0.2 mm. Right panels: Diagrams showing take rate and fat pad filling. (**B**) Representative images of carmine-stained outgrowths obtained 5 weeks after transplantation of intact secondary spheres harvested from untreated and HGF-treated cultures of luminal progenitors. Take rates are indicated. Bar, 2 mm. (**C**) Enlarged view of the outgrowth shown in **B**. Bar, 0.8 mm. (**D**) Double immunofluorescence stainings of sections through the outgrowth shown in **C**. Immunodetection of K5/K8 in a duct (left panel) and α-SMA/TK in a duct and a growing bud (middle and right panels, respectively). Bar, 45 μm. The following supplementary file is available for *Figure 7*: *Supplementary file 1*.

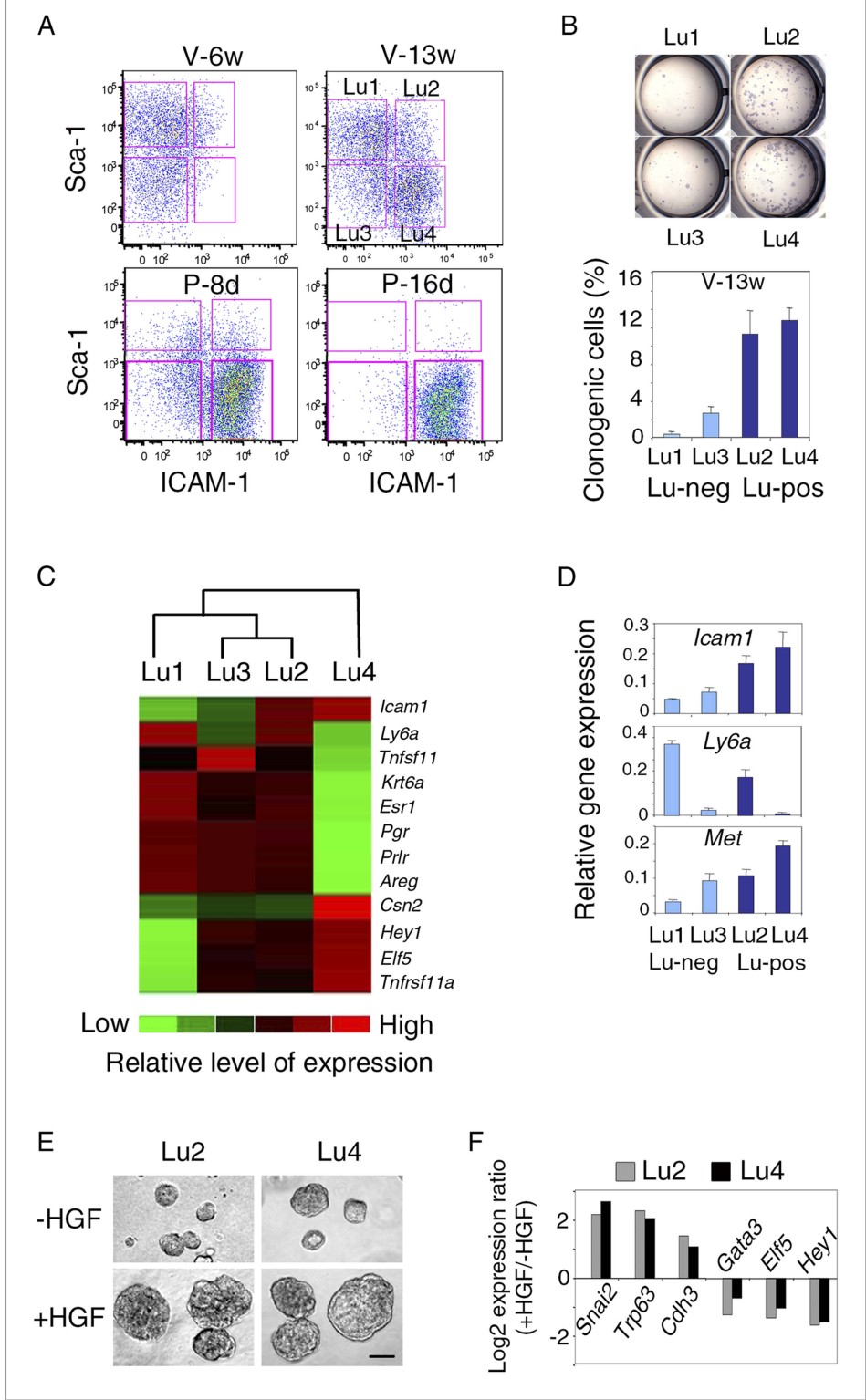

**Figure 8**. Met-expressing clonogenic cells are distributed in hormone-receptor-positive and negative luminal cell populations. (**A**) Representative flow cytometry dot-plots showing Sca-1 and ICAM-1 expression in the mammary luminal cell population isolated from mice at different stages of development: V-6w, 6-week-old virgin; V-13w, 13-week-old virgin; P-8d, 8-day pregnant; P-16d, 16-day pregnant mice. Combined with Sca-1, ICAM-1 discriminated four cell populations referred to as Lu1, Lu2, Lu3, and Lu4. (**B**) Colony formation by Lu1, Lu2, Lu3, and Lu4 cell subsets isolated from mammary glands of mature virgin mice. Upper panel: H&E staining of colonies after 8 days in *Figure 8. continued on next page*

*Figure 8. Continued*
culture. Lower panel: percentages of clonogenic cells. The results from two independent cell samples (each of which with three separate wells) are presented as mean values ±S.E.M. (**C**) Heat map of qPCR gene expression analysis performed on Lu1, Lu2, Lu3, and Lu4 cells freshly isolated from mammary glands of mature virgin mice. The qPCR values were normalized to *Gapdh* expression. Mean values from three independent cell preparations were used to establish the map and determine relationships between the luminal subsets by unsupervised hierarchical clustering. (**D**) q-PCR analysis of *Icam1*, *Ly6a*, and *Met* expression in Lu1, Lu2, Lu3, and Lu4 cell populations freshly isolated from mammary glands of mature virgin mice. The values were normalized to *Gapdh* expression and represent mean values ±S.E.M from three independent preparations. (**E**) Representative phase contrast images of spheres derived from Lu2 and Lu4 cell populations grown in the absence or presence of HGF for 10 days. Bar, 150 μm.
(**F**) Comparative expression levels of basal-specific, EMT-associated and luminal-specific genes in spheres derived from Lu2 and Lu4 cells, as determined by q-PCR. Cells were grown as described above in (**E**). Results are expressed as Log2 ratios of values normalized to *Gapdh*. The comparator values were those obtained with cell preparations grown in the absence of HGF.
The following source data and figure supplement are available for figure 8:
**Source data 1**.
**Figure supplement 1**. Characterization of luminal cell populations isolated from virgin and pregnant mice using ICAM-1 and Sca-1 expression.

All four luminal cell subsets expressed K8 and displayed similar high levels of *Krt18* and very low levels of the basal-specific gene *Krt14* (*Figure 8A*, *Figure 8—figure supplement 1B*). We characterized the four cell subsets further by carrying out qPCR-based gene expression analyses on a panel of luminal-specific genes. Unsupervised hierarchical clustering divided the populations into two main branches (*Figure 8C*). Lu1, Lu2, and Lu3 clustered together, sharing strong expression of hormone receptors, ERa, PR, and PrlR. The second branch contained only Lu4, which displayed very low levels of hormone receptor transcripts.

Only Lu4, the major clonogenic subset devoid of HR-positive cells, expressed the milk protein β-casein (*Figure 8C*). It also strongly expressed *Elf5* and *Tnfrsf11a*, two crucial regulators of alveologenesis (*Fata et al., 2000*; *Oakes et al., 2008*). Interestingly, flow cytometry analysis on mammary epithelial cells isolated at different stages of development showed that Lu4 was absent at puberty; this subset appeared in the glands of sexually mature cycling females and displayed massive expansion early in pregnancy (*Figure 8A*). By contrast, Lu2 and Lu3 were detected at all stages of development. As in mature virgin glands, Lu2 and Lu4 were highly clonogenic during early pregnancy, whereas Lu3 was not (*Figure 8—figure supplement 1C*). The gene expression patterns of Lu1, Lu2, Lu3, and Lu4 cells isolated from mature virgin and early pregnant mice were similar (*Figure 8—figure supplement 1D,E*).

The q-PCR analysis showed that *Met* was strongly expressed in the clonogenic populations, Lu2 and Lu4 (*Figure 8D*). These populations responded similarly to HGF treatment in 3D cultures. On stimulation, both produced larger spheres, displaying an upregulation of basal-specific genes (*Snai2*, *Trp63*, and *Cdh3*) and a down-regulation of luminal-specific regulatory genes (*Gata3*, *Elf5*, and *Hey1*) (*Figure 8E,F*).

These data show that luminal progenitors defined by ICAM-1 expression include hormone receptor-positive and hormone receptor-negative cells, all of which are potential targets of HGF/Met signaling.

## ICAM-1 can serve to localize luminal progenitors in the mammary epithelium

To complement the flow cytometry data, we analyzed ICAM-1 localization on histological sections of mammary gland. Immunohistochemical studies confirmed that in the pubertal gland, basal myoepithelial rather than luminal cells expressed ICAM-1 while at late pregnancy, the whole epithelium was positive (*Figure 9A*). At the onset of lactation, ICAM-1 was down-regulated both in ducts and alveoli (*Figure 9A*).

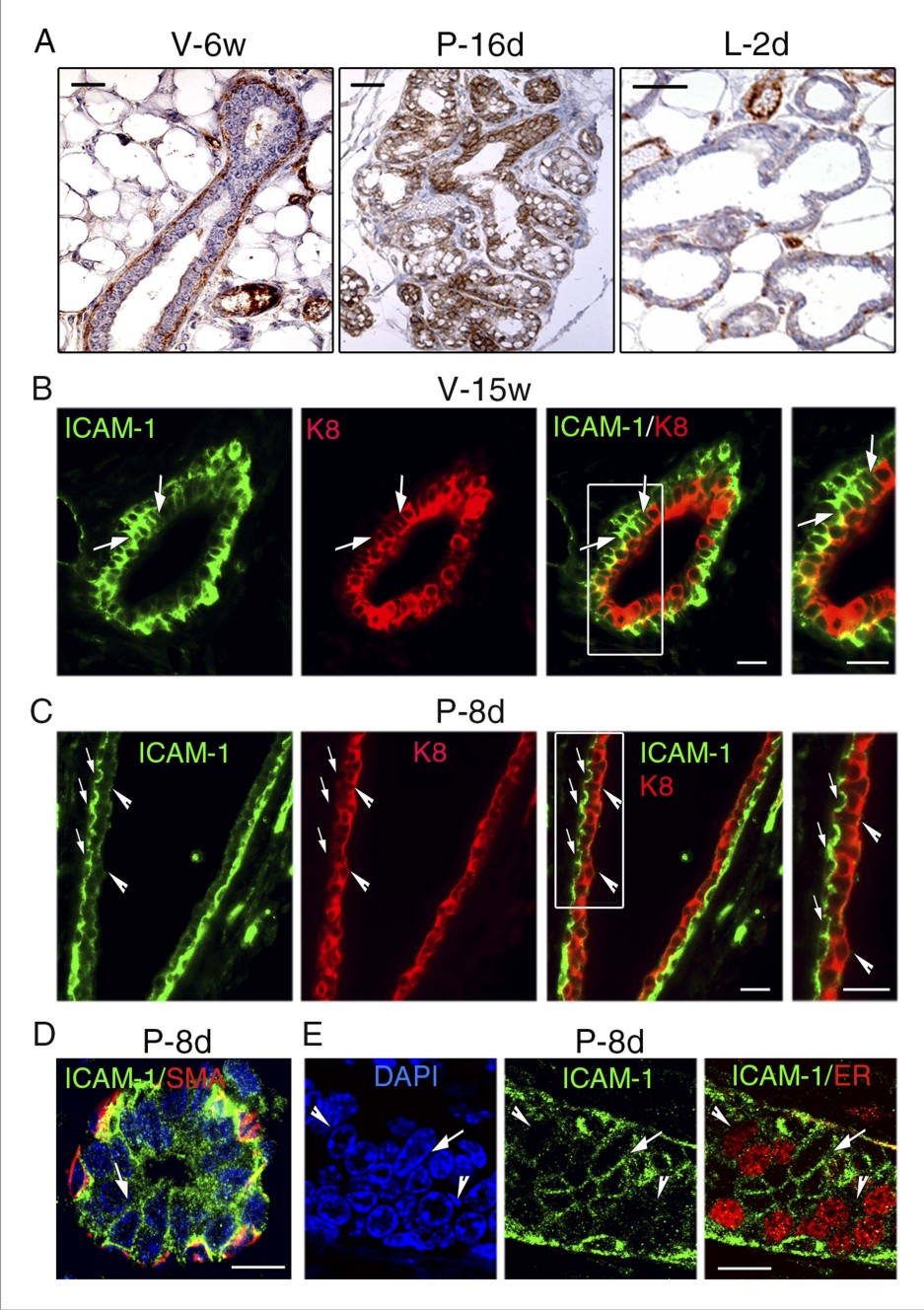

**Figure 9**. ICAM-1 localization in the developing and adult mammary gland. (**A**) Immunohistochemical detection of ICAM-1 in sections through mammary glands from mice at puberty (V-6w; left panel), late pregnancy (P-16d; middle panel), and onset of lactation (L-2d; right panel). Left panel: bar, 60 μm. Middle and right panels: bar, 100 μm. (**B**) Double immunofluorescence labeling of ICAM-1 and K8 in a mammary duct of a 15-week-old virgin female (V-15w). The right panel shows an enlarged view of the area defined on the left. The arrows indicate a cluster of ICAM1-positive luminal cells. Bars, 40 μm. (**C**) Double immunofluorescence staining of ICAM-1 and K8 in a large mammary duct at early gestation (P-8d). The right panel shows enlarged view of the area defined on the left. The arrows point to myoepithelial cells expressing ICAM-1 at their apical and lateral surfaces. The arrowheads indicate luminal layer negative for ICAM-1. Bars, 40 μm. (**D** and **E**) Sections through an alveolus and a small duct from a mammary gland of 8-day-pregnant mouse (P-8d). (**D**) Double immunofluorescence labeling of ICAM-1 and α-SMA (SMA); (**E**) ICAM-1 and ER. DAPI served to stain the nuclei. The arrows point to clusters of ICAM1-positive cells. The arrowheads indicate clusters of ER-positive cells negative for ICAM-1. Bars, 10 μm.

*Figure 9. continued on next page*

*Figure 9. Continued*

The following figure supplement is available for figure 9:

**Figure supplement 1**. Response of luminal progenitors isolated from *Icam1*-deficient mammary epithelium to HGF stimulation.

ICAM-1 localization in the adult mammary epithelium was further investigated using double immunofluorescence labeling with anti-ICAM-1 and anti-K8 antibodies. In agreement with the flow cytometry data, mammary basal cells in adult virgin mice displayed high levels of ICAM-1, whereas the luminal compartment comprised ICAM1-negative and ICAM1-positive cells (*Figure 9B*). ICAM1-positive luminal cells, enriched in clonogenic cells, often formed cell clusters within the ducts.

At early pregnancy, myoepithelial cells lining large ducts strongly expressed ICAM-1, whereas luminal cells were devoid of ICAM-1 (*Figure 9C*). In contrast, in the developing alveoli and small ducts, the luminal layer, presumably enriched in alveolar progenitors, contained clusters of cells positive for ICAM-1 (*Figure 9D*). Consistently with qPCR data, ER-positive cells were mostly ICAM1-negative (*Figure 9E*). Notably, in adult virgin and early pregnant mice, ICAM-1 expression was restricted to cell–cell contacts, basal-to-basal, basal-to-luminal, and luminal-to-luminal (*Figure 9B–E*).

## Loss of ICAM-1 does not impede the response of luminal progenitors to HGF

To analyze a possible contribution of ICAM-1 in mammary development and HGF/Met signaling, we investigated the mammary phenotype of *Icam1*-deficient (*Icam1*-KO) mice. The *Icam1*-KO females are viable, fertile, and able to feed their pups (*Xu et al., 1994*). Whole-mount analysis of mammary glands from wild-type and *Icam1*-KO adult virgin and pregnant mice indicated that lack of ICAM-1 did not induce visible defects in mammary morphogenesis (*Figure 9—figure supplement 1A*). Flow cytometry analysis showed that the percentages of basal and luminal cells were not significantly affected in the *Icam1*-KO epithelium of adult virgin mice (*Figure 9—figure supplement 1B*).

For HGF stimulation assays, we isolated *Icam1*-KO mammary epithelial cells and used the Sca-1-negative luminal cell population enriched in clonogenic progenitors (*Figure 9—figure supplement 1C*). We found that *Icam1*-KO Sca-1-negative luminal cells responded to HGF by increased clonogenicity and production of large spheres (*Figure 9—figure supplement 1D,E*). In addition, as previously observed in the presence of ICAM-1, HGF treatment led to the induction of the basal-specific genes *Krt5*, *Trp63*, and *Snai2*, with a concomitant decrease in the regulators of the luminal lineage, *Elf5*, *Gata3*, and *Hey1* (*Figure 9—figure supplement 1F,G*). Thus, ICAM-1 is not mandatory for mammary development and HGF/Met signaling in luminal progenitors.

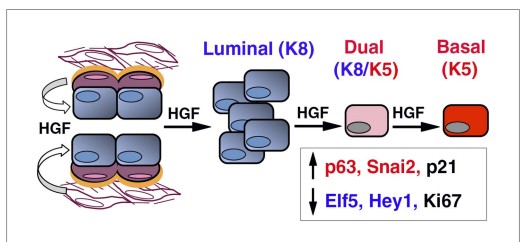

**Figure 10**. HGF/Met signaling in the mammary tissue. Luminal progenitors express Met, whereas stromal and basal myoepithelial cells produce HGF. Paracrine HGF/Met signaling can control the survival and proliferation of luminal progenitors and affects their fate by modulating antagonistic regulators of basal- (p63, Snail2) and luminal-specific (Elf5, Hey1) regulators.

## Discussion

We report here that ICAM-1 is a new surface marker for the enrichment of mouse mammary luminal progenitors. Luminal progenitor population identified by ICAM-1 labeling is heterogeneous, comprising HR-positive and HR-negative cells, potentially ductal and alveolar progenitors, both strongly expressing Met. We show that paracrine Met activation stimulates the clonogenic activity of ICAM1-expressing luminal progenitors, controlling their survival and proliferation, and promotes a luminal-to-basal switch while triggering EMT program (*Figure 10*). EMT was accompanied by acquisition of regenerative potential, a property restricted to multipotent stem cells. Our data suggest that paracrine HGF signaling can affect the fate of luminal progenitors during mammary development and tumorigenesis.

## ICAM-1 is a new marker for the enrichment of clonogenic luminal progenitors

Our study provides the first evidence showing that ICAM-1 is differentially distributed, both spatially and temporally, in the mouse mammary epithelium. In mature virgin mice and early in pregnancy, the luminal cell population comprises ICAM1-negative and ICAM1-positive cell fractions, highly enriched in hormone-sensing cells and clonogenic progenitors, respectively. The luminal progenitors identified by ICAM-1 staining strongly express *Elf5*, *Hey1*, *Tnfrsf11a*, and *Rspo1* genes encoding critical regulators of mammary development (*Fata et al., 2000*; *Bouras et al., 2008*; *Lee et al., 2011*; *Cai et al., 2014*).

Several recent studies have indicated that the luminal progenitor population includes cells with different phenotypes (*Oliver et al., 2012*; *Regan et al., 2012*; *Shehata et al., 2012*; *Lafkas et al., 2013*; *Sale et al., 2013*; *Rodilla et al., 2015*). The differential expression of Sca-1 and CD61 was first used to obtain partial enrichment in luminal progenitor cells (*Asselin-Labat et al., 2007*; *Sleeman et al., 2007*; *Rios et al., 2014*). Subsequently, the combined use of Sca-1 and c-kit or CD49b revealed that the luminal progenitors in the mature virgin gland belong to two distinct populations, one consisting essentially of HR-positive cells and the other mostly of HR-negative cells (*Regan et al., 2012*; *Shehata et al., 2012*). We found that in virgin glands and during early pregnancy, combined ICAM-1 and Sca-1 expression discriminated these two major luminal clonogenic cell subsets. ICAM-1 clearly appeared more efficient than Sca-1 for separating non-clonogenic and clonogenic luminal populations. Importantly, ICAM-1 marked the luminal progenitor population in different mouse strains, including C57Bl6/J, and in mixed backgrounds, such as that of BlgCre; R26 mice, whereas neither CD61 nor c-kit can be used for luminal progenitor enrichment from C57Bl6/J mouse mammary glands (*Visvader and Stingl, 2014*). Furthermore, differential ICAM-1 expression within the Sca-1-negative luminal cell population separated a poorly clonogenic ICAM1-negative subset enriched in HR-positive cells (Lu3) from a major clonogenic cell fraction positive for ICAM-1 and lacking HR expression (Lu4). Such separation was not possible with CD49b, the most recently described marker for characterizing luminal cell subsets (*Shehata et al., 2012*; *De Silva et al., 2015*).

The clonogenic luminal cell population is thought to comprise distinct ductal- and alveolar-restricted progenitors, as yet not fully characterized (*Regan et al., 2012*; *Shehata et al., 2012*). Cells committed to the alveolar fate are expected to expand in adult females particularly at the onset of pregnancy, to express *Tnfrsf11a* and *Elf5* strongly, and most probably, to lack ER and PR (*Fata et al., 2000*; *Lee et al., 2011*; *Fu et al., 2014*; *Rodilla et al., 2015*). Interestingly, the Sca-1-negative ICAM1-positive clonogenic cell population (Lu4) absent from prepubertal mammary epithelium fulfills these criteria. In addition, these cells have high levels of transcript for the milk protein β-casein. By contrast, the Sca-1- and ICAM1-positive clonogenic subset containing HR-expressing cells (Lu2) present throughout mammary development may preferentially contain ductal-restricted progenitors.

Thus, ICAM-1 is a new robust marker for separating basal and luminal cells and analyzing the heterogeneity of the luminal cell compartment in the adult mammary gland using flow cytometry. In addition, ICAM-1 can be used to localize and further characterize luminal progenitors in situ.

## HR-negative and HR-positive luminal progenitors are potential targets of HGF/Met signaling

The target cells for Met signaling in the mammary epithelium remain poorly characterized. A recent work reported the expression of Met predominantly in the Sca-1-negative luminal progenitor-enriched population (*Gastaldi et al., 2013*). In agreement with these data, we found high-Met transcript levels in the Sca-1-negative ICAM1-positive clonogenic cell population (Lu4). Additionally, we observed strong Met expression in the Sca-1-positive ICAM1-positive luminal progenitors (Lu2). Thus, the two major clonogenic populations of luminal progenitors, enriched in HR-negative and HR-positive cells, are potential targets of HGF/Met signaling. Met was also expressed in the Sca-1-negative ICAM1-negative cell subset (Lu3), which was weakly clonogenic and enriched in HR-positive cells. Our data thus highlight the phenotypic diversity of Met-expressing cells in the luminal population. Consistently, a knockin mouse model with a mutationally activated Met has been reported to develop various types of mammary tumors, including ER-positive and ER-negative tumors (*Graveel et al., 2009*).

We found that ICAM1-expressing luminal progenitors responded to HGF stimulation by basal marker upregulation. Importantly, using the Blg-Cre; R26 reporter mouse, we demonstrated that these basal marker-expressing cells were of luminal origin. Triple-negative basal-like breast cancers are thought to originate from luminal progenitors aberrantly expressing basal-specific markers (*Lim et al., 2009*; *Molyneux et al., 2010*; *Proia et al., 2011*). The highest levels of Met have been associated with the basal subtype of breast cancers (*Garcia et al., 2007*; *Graveel et al., 2009*; *Ponzo et al., 2009*; *Gastaldi et al., 2010*; *Knight et al., 2013*). Moreover, amplifications of the Met locus and Met overexpression occur in Brca1/p53-deficient mouse tumors, a model of basal-like breast cancers (*Smolen et al., 2006*; *Gastaldi et al., 2013*). Thus, consistent with these reports, our results strongly indicate that Met signaling may contribute to the phenotypic characteristics of certain basal-like breast carcinomas.

## ICAM-1 is not mandatory for HGF/Met signaling in luminal progenitors

We found that in the mouse mammary epithelium, Met-expressing luminal progenitors displayed both ICAM-1 and CD44v6 at their surface. In mouse hepatocytes, CD44v6 acts as main co-receptor for Met, however, ICAM-1 can trigger Met activation in the absence of CD44 (*Orian-Rousseau et al., 2002*; *Olaku et al., 2011*). Our data showed that loss of ICAM-1 did not severely affect mammary development, HGF/Met signaling, and basal cell differentiation process in luminal progenitors. This suggests that CD44v6 rather than ICAM-1 serve as main co-receptor for Met in the mouse mammary epithelium and that identical compensation mechanisms may be involved in Met activation in liver and mammary gland. Distribution of Met and its co-receptors in the human mammary gland remains poorly defined; however, strong expression of Met, ICAM-1, and CD44v6 has been associated with triple-negative breast cancers and correlated with poor prognosis (*Charpin et al., 2009*; *Gastaldi et al., 2010*; *Ponzo and Park, 2010*; *Schroder et al., 2011*). Moreover, ICAM-1 has recently been identified as a molecular target for triple-negative breast cancers (*Guo et al., 2014*).

## Myoepithelial and stromal cells may both control luminal progenitor function in a paracrine manner

Stromal cells are usually considered to be the only source of HGF in the mammary gland (*Niranjan et al., 1995*; *Yang et al., 1995*; *Gastaldi et al., 2013*). However, we show here that mammary myoepithelial cells also produce HGF. It is worth mentioning that in the postnatal mammary epithelium, ductal luminal cells are in direct contact with myoepithelial rather than stromal cells. In alveoli, the myoepithelial cell layer is discontinuous, so luminal cells may also come into contact with the basement membrane (*Moumen et al., 2011*). HGF produced by both myoepithelial and stromal cells may therefore contribute to the control of the luminal progenitor activity, favoring either their normal expansion during development or aberrant amplification in cancers. Paracrine basal-to-luminal signaling mediated by the Nrg1-Erbb4 ligand–receptor pair has recently been implicated in the regulation of luminal cell maturation during lobulo–alveolar development (*Forster et al., 2014*).

We found that HGF had multiple effects on Met-expressing luminal progenitors. Early in culture, HGF promoted cell survival and proliferation. Consistent with its role in tubulogenesis (*Rosario and Birchmeier, 2003*), HGF then favored lumen formation. Finally, HGF treatment affected the fate of luminal progenitors, leading to the accumulation of a cell population expressing basal markers, including cells co-expressing basal and luminal keratins. Such dual luminal/basal cells are seldom in the adult mammary epithelium but are abundant in fetal mammary rudiments (*Spike et al., 2012*). The accumulation of cells co-expressing basal and luminal keratins has been observed in the mammary epithelium of Elf5-null mice, which have high-Snail2 levels and display EMT-like phenotypic changes (*Chakrabarti et al., 2012a*, *2012b*).

## HGF triggers EMT in luminal progenitors and modulates the expression of antagonistic regulators of luminal and basal fates

Several regulators of the balance between mammary luminal and basal phenotypes have recently been identified. Notch signaling specifies a luminal cell fate, whereas ΔNp63 is required for the maintenance of basal cell characteristics (*Bouras et al., 2008*; *Yalcin-Ozuysal et al., 2010*). Forced expression of Snail2 endows mammary luminal progenitor cells with molecular characteristics and functional properties of basal-type stem cells, whereas Elf5, through the direct repression of *Snai2*, promotes luminal cell differentiation (*Guo et al., 2012*; *Chakrabarti et al., 2012a*, *2012b*). Consistent with these data, we found that HGF induced the expression of *Trp63* whilst repressing that of *Hey1*,

a major Notch effector, in luminal progenitors. Concomitantly, we observed important hallmarks of EMT, including an increase in *Snai1* and *Snai2* levels, *Elf5* repression and claudin down-regulation. The transcription of P-cadherin and N-cadherin was induced upon HGF treatment, but E-cadherin expression was not repressed, suggesting a partial, reversible EMT process (*Nieto and Cano, 2012*; *Gonzalez and Medici, 2014*). Consistently, alterations of the luminal progenitor phenotype did not persist in vitro after HGF withdrawal. Altogether, these data suggest that microenvironmental signals, such as HGF, may modulate the phenotype of luminal progenitors, by affecting the expression of antagonistic regulators of the basal and luminal fates.

In addition to their role in triggering EMT, transcription factors of the Snail family have been reported to attenuate cell cycle progression by increasing p21 expression (*Vega et al., 2004*). At late time points, *Cdkn1a*, *Snai1*, and *Snai2* were upregulated in HGF-treated cultures of luminal progenitors, whereas *Mki67* was down-regulated. These changes were accompanied by the accumulation of K5-expressing quiescent cells. How p21 and its major upstream regulator p53 are involved in the response of luminal progenitors to HGF remains to be unraveled. A recent analysis of transgenic mouse models showed that Met can synergize with p53 loss to promote the formation of triple-negative mammary tumors with a claudin-low, EMT-type molecular signature (*Knight et al., 2013*). Furthermore, the loss of p53 from the mammary epithelium leads to the expansion of clonogenic mammary luminal progenitors (*Chiche et al., 2013*).

In brief, our data provide new insights into the cellular and molecular mechanisms underlying the phenotypic plasticity of luminal progenitors. They also suggest that mammary basal and stromal cells may both be involved in controlling luminal progenitor function via the paracrine activation of Met during mammary development and tumorigenesis.

## Materials and methods

### Mouse strains and transgenic mice

BALB/cByJ JAX, C57BL/6, and BALB/c-Nude JAX females were purchased from Charles Rivers (L'arbresle, France). Transgenic mice expressing the Cre recombinase under the control of the β-Lactoglobulin promoter (Blg-Cre) were from the Jackson Laboratories (Sacramento, CA, USA) and purchased from Charles Rivers. Rosa26-LacZ reporter strain was provided by *Soriano (1999)*. *Icam1*-deficient mice (B6.129S4-*Icam1*$^{tm1Jcgr}$/J) were from the Jackson Laboratories. The mature virgin females used in the experiments were 11–25 weeks old. The care and use of animals used here was strictly applying European and National Regulation for the Protection of Vertebrate Animals used for Experimental and other Scientific Purposes in force (facility licence #C75-05-18). It complies also with internationally established principles of replacement, reduction, and refinement in accordance with Guide for the Care and Use of Laboratory animals (NRC, 2011). All experimental procedures were ethically approved (Ethical approval 02265.01/CEEA-IC 118).

### Preparation of mammary epithelial cells

Single cells were prepared from 2 to 8 inguinal mammary glands. Glands taken from virgin or pregnant females were minced, first with scissors and then with scalpels. Minced tissues were transferred to a digestion solution containing 3 mg/ml collagenase (Roche Diagnostics, Meylan, France), 100 units/ml hyaluronidase (Sigma–Aldrich, Saint Louis, MO, USA) in $CO_2$-independent medium (Gibco, Life Technologies, St Aubin, France) completed with 5% fetal bovine serum (FBS, Lonza, Basel, Switzerland) and 2 mM L-glutamine (Sigma–Aldrich), and incubated for 90 min at 37°C with shaking (150 rpm). Digested samples were centrifuged at 450×*g* for 5 min and the supernatant eliminated. Pellets were washed once with $CO_2$-independent medium, treated with a prewarmed 0.25% trypsin/0.1% versen (Life Technologies; Biochrom, Berlin, Germany) solution in Phosphate Buffered Saline (PBS) for 1 min and rinsed with $CO_2$-independent medium containing 5% FBS. Pellets were then resuspended in a solution of 5 mg/ml dispase II (Roche) in $CO_2$-independent medium containing 5% FBS and DNAseI (Sigma–Aldrich) was added to a final concentration of 0.1 mg/ml. After 5 min incubation at 37°C, cells were rinsed once in $CO_2$-independent medium containing 5% FBS and the pellets were treated with a cold ammonium chloride solution (Stem Cell Technologies, Grenoble, France). Cell suspensions were centrifuged, resuspended in $CO_2$-independent medium, and filtered through a nylon mesh cell strainer with 40-μm pores (Fisher Scientific, Illkirch, France) before immunolabeling.

## Flow cytometry

Freshly isolated cells were incubated at 4°C for 20 min with the following conjugated antibodies: anti-CD24-FITC (clone M1/69) and anti-CD49f-PE (clone GoH3) from BD Biosciences (BD France, Le Pont-de-Claix, France), anti-CD54-PE (clone YN1/1.7.4), anti-CD45-APC (clone 30-F11), and anti-CD31-APC (clone MEC13.3) from Biolegend (San Diego, CA, USA), anti-Ly6A/E-PE-Cy5 (clone D7) from eBiosciences (San Diego, CA, USA). Labeled cells were sorted on a FACSVantage flow cytometer (BD Biosciences, San Jose, CA, USA), and data analyzed using FlowJo software. Sorted cell populations were routinely re-analyzed and found to be 96–98% pure. As estimated by trypan blue exclusion, cell viability after sorting was between 83% and 92%.

## Cell culture assays

For two-dimensional clonogenic assays, sorted luminal cells were plated on irradiated 3T3 cell feeders on 24-well plates at a density of 250–500 cells per well and cultured in DMEM/F12 medium supplemented with 10% FBS, 5 μg/ml insulin (Sigma–Aldrich), 10 ng/ml EGF (Invitrogen, Life Technologies), and 100 ng/ml cholera toxin (ICN Biochemicals, Irvine, CA, USA) for 7 days as described elsewhere (*Sleeman et al., 2007*). For mammosphere cultures, freshly isolated luminal cells were seeded on ultra-low adherence 24-well plates (Corning, NY, USA) at the density of 2000–5000 cells/well, in mammosphere media: DMEM/F12 medium supplemented with 2% B27, 20 ng/ml EGF, 20 ng/ml bFGF (Gibco, Life Technologies), 4 μg/ml heparin (Sigma–Aldrich), 10 μg/ml insulin, and 2% growth-factor-reduced Matrigel (BD Biosciences) as described elsewhere (*Spike et al., 2012*; *Chiche et al., 2013*). When specified, mammospheres were treated with 25–50 ng/ml recombinant mouse HGF (R&D Systems Europe, Lille, France) every 2 days. ImageJ software was used to count the colonies and the mammospheres and evaluate their size in pixels. Single cell suspensions were obtained from mammospheres by treatment with 0.05% trypsin (Gibco, Life Technologies).

Caspase-3/7 activity was assessed in cells harvested 1 and 2 days after plating them in mammosphere medium, using the Caspase-Glo 3/7 Assay System (Promega, Madison, WI, USA) according to the manufacturer's protocol.

## Immunofluorescence and immunohistochemical stainings

Freshly isolated cells were cytocentrifuged onto slides and fixed in cold methanol for 10 min. Cells were incubated with primary antibodies at 37°C for 1 hr, with secondary antibodies at room temperature for 1 hr and mounted in Prolong Gold antifade reagent with DAPI (Invitrogen, Life Technologies). Mammospheres were resuspended in 50 μl Matrigel and incubated at 37°C for 2 hr prior to embedding either in OCT (Tissue-Tek, Sakura Finetek Europe, Leiden, Netherlands) for cryosections or in paraffin. Cryosections were post-fixed with 4% paraformaldehyde for 10 min and treated with 0.5% triton X-100 for 5 min before immunostaining. Spheres were fixed in Methacarn (1:3:6 mixture of acetic acid:chloroform:methanol) before embedding in paraffin. Sections were dewaxed, processed for acidic antigen retrieval, and immunolabeled, as described elsewhere (*Chiche et al., 2013*). For cell proliferation assays, mammospheres were incubated with BrdU (5 μg/ml; Sigma–Aldrich) for 1 hr before being fixed and processed.

The following primary antibodies were used: anti-K5 and anti-K8 (Covance, Princeton, NJ, USA), anti-BrdU, anti-CD44v6 (AbD Serotec, Oxford, UK), anti-E-cadherin (ECCD2; Life Technologies), anti-p63 and anti-claudin-1 (Abcam, Cambridge, UK), anti-snail2/slug (Cell Signaling Technology, Danvers, MA, USA), anti-ICAM-1 (Proteintech, Chicago, IL, USA), anti-ER (Dako France, Les Ulis, France), and Cy3-conjugated anti-α-SMA (Sigma–Aldrich). Alexafluor-conjugated secondary antibodies (Molecular Probes, Life Technologies) were used. For the immunohistochemical detection of ICAM-1, we used the EnVision System from Dako. Image acquisition was performed using a Leica DM 6000B microscope (Wetzlar, Germany) and MetaMorph software. Confocal images were acquired with a Nikon Confocal A1r microscope using a 60× CFI Plan oil objective (Apo VC/NA 1.4/WD 0.13).

## X-gal staining

Mammospheres established from Blg-Cre; R26 mammary cells were dissociated and isolated cells were pre-fixed with 2.5% paraformaldehyde for 5 min at 4°C and X-gal stained in suspension at 37°C

overnight. Cells were cytocentrifuged, post-fixed with 4% paraformaldehyde for 10 min at room temperature, and processed either for Fast Red staining or immunostaining.

For whole-mount X-gal staining, mammary glands from Blg-Cre; R26 females were fixed in 2.5% paraformaldehyde for 1 hr at 4°C and stained overnight at 30°C. Glands were embedded in paraffin and sections counterstained with Fast Red.

## Transplantation assays and whole-mount analysis

Isolated cells derived from primary spheres or intact secondary spheres were transplanted into the inguinal fat pads of 3-week-old BALB/c-Nude JAX females cleared of endogenous epithelium as described elsewhere (*Deome et al., 1959*; *Moumen et al., 2012*; *Chiche et al., 2013*). Untreated and HGF-treated cells or spheres were resuspended in 10 µl of 25% growth factor-reduced Matrigel before being grafted in the contralateral fat pads of host mice. For secondary sphere transplantations, 5000 ICAM1-positive luminal progenitors per well were plated, grown with or without HGF for 14 days, then, dissociated and replated at 5000 cells per well as for primary spheres. After 15 days in culture, the content of each culture well was grafted in a separate mammary fat pad. To analyze the outgrowths, dissected mammary fat pads were spread onto glass slides, fixed in Methacarn, and stained with carmine alum (Stem Cell Technologies), as described elsewhere (*Moumen et al., 2012*; *Chiche et al., 2013*).

## Reverse transcription-polymerase chain reaction

RNA was reverse-transcribed with MMLV H(−) Point reverse transcriptase (Promega, Madison, WI, USA), and quantitative PCR (qPCR) was performed by monitoring, in real time, the increase in fluorescence of the SYBR Green dye on an LightCycler 480 Real-Time PCR System (Roche Applied Science, Basel, Switzerland). The primers used for qPCR analysis were purchased from SABiosciences/ Qiagen (Hilden, Germany) or designed using Oligo 6.8 software and synthesized by Eurogentec (Seraing, Belgium) (*Supplementary file 2*).

## Statistical analysis of the data

p values were determined using Student's test with two-tailed distribution and unequal variance.

## Acknowledgements

We are particularly grateful to the personnel of the Animal facility (Sonia Jannet, Isabelle Grandjean) and the Flow Cytometry Core facility (Annick Viguier, Sophie Grondin and Zosia Maciorowski). We greatly thank the staff of the PICT-IBiSA Lhomond Imaging facility and the Nikon Imaging Center at Institut Curie–Centre National de la Recherche Scientifique for help with image acquisition, in particular Lucie Sengmanivong. We also thank Veronica Rodilla (Institut Curie, Paris, France) for sharing reagents and protocols. Laura Bresson received funding from the « *Ministère de la Recherche et de l'Enseignement Supérieur* ».

## Additional information

### Funding

| Funder | Grant reference | Author |
|---|---|---|
| Ligue Contre le Cancer | Equipe Labelisee 2013 | Marina A Glukhova |
| Agence Nationale de la Recherche | ANR-13-BSV2-0001 | Marina A Glukhova |
| Canceropole Ile de France | 2014-1-SEIN-01-ICR-1 | Marina A Glukhova |
| Agence Nationale de la Recherche | ANR-08-BLAN-0078-01 | Marina A Glukhova |

The funders had no role in study design, data collection and interpretation, or the decision to submit the work for publication.

### Author contributions

AD-C, VP, AC, LB, MR, DM, Acquisition of data, Analysis and interpretation of data; VO-R, MMV, Analysis and interpretation of data, Contributed unpublished essential data or reagents; MMF, Conception and design, Analysis and interpretation of data; MAG, Analysis and interpretation of

data, Drafting or revising the article; M-AD, Conception and design, Acquisition of data, Analysis and interpretation of data, Drafting or revising the article

## Ethics

Animal experimentation: The care and use of animals used here was strictly applying European and National Regulation for the Protection of Vertebrate Animals used for Experimental and other Scientific Purposes in force (facility licence #C75-05-18). It complies also with internationally established principles of replacement, reduction and refinement in accordance with Guide for the Care and Use of Laboratory animals (NRC 2011). All experimental procedures were ethically approved (Ethical approval 02265.01/CEEA-IC 118).

## Additional files

### Supplementary files

• Supplementary file 1.  Regenerative capacity of cells isolated from untreated and HGF-treated primary spheres.

• Supplementary file 2.  List of primers for q-PCR.

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
