## [Decision Letter]

Thank you for sending your work entitled “Paracrine Met signaling triggers epithelial-mesenchymal transition in mammary luminal progenitors, affecting their fate” for consideration at *eLife*. Your article has been favorably evaluated by Fiona Watt (Senior editor) and three reviewers, one of whom is a member of our Board of Reviewing Editors.

The Reviewing editor and the other reviewers discussed their comments before we reached this decision, and the Reviewing editor has assembled the following comments to help you prepare a revised submission.

The general consensus is that this paper is in principle appropriate for *eLife* as it identifies ICAM-1 as a novel marker to discriminate between HR-positive and negative mammary gland luminal clonogenic subsets and basal cells and this finding has the potential to be of significant general interest. However, as indicated by the reviewers there are some major concerns and significant ambiguities that need to be addressed before the paper can be considered for resubmission.

1) Most of the experiments, and thus the majority of the conclusions, have been based on in vitro data including flow cytometry and various cell culture assays. The in vivo role of ICAM-1 luminal progenitor cells has only been partially addressed using a fat pad reconstitution experiment that demonstrates the dependence of ICAM-1 positive luminal progenitor cells on HGF stimulation. This one in vivo experiment is potentially very important and should be expanded upon to show some functional role of ICAM-1 in mammary gland development (Figure 3–figure supplement 5). Firstly, the number of experimental mice should be increased to obtain a meaningful statistical result. These results should then be incorporated into a main figure. Moreover, some IHC analysis to show the localisation of ICAM-1 luminal progenitor cells during mammary gland development would be useful.

2) Previous work published by [19] has demonstrated the role of MET and HGF in the luminal progenitor cell function in vivo, albeit not specifically for the ICAM-1 positive subpopulation. Since the novelty of this manuscript relies on the identification of ICAM-1 as a marker of luminal progenitor cells and as the generation of basal-like cells is dependent on c-Met, it is important to consolidate the functional contribution of ICAM-1 to this process. Is ICAM-1 necessary for EMT/the transition of progenitor cells to basal like cells in response to HGF? Furthermore, based on the knowledge that ICAM-1 is a co-receptor of c-Met (reported in an independent publication by one of the co-authors), the novelty of the report could be substantially expanded by the assessment whether ICAM-1 also functionally contributes to the basal cell differentiation process. In vivo experiments addressing directly the requirement for ICAM-1 are therefore necessary to address this point.

3) There are also some specific ambiguities with regard to Figure 4 and Figure 4, specifically in the E-cadherin staining. In Figure 4, E-cadherin-mediated cell-cell adhesion does not seem to change with and without HGF-stimulation? In Figure 4, the absence of surface E-cadherin staining in K5-positive cells is not obvious. Better resolution microscopy images should be provided. Claudin-I immunofluorescence staining may resolve the unclear picture of the maintenance/loss of cell adhesion. It would also be helpful to validate the EMT-like changes in gene expression on the protein levels for example using immunofluorescence imaging.

4) Several experiments lack statistical analysis and these should be addressed to strengthen the conclusions being made.

---

## [Author Response]

*1) Most of the experiments, and thus the majority of the conclusions, have been based on in vitro data including flow cytometry and various cell culture assays. The in vivo role of ICAM-1 luminal progenitor cells has only been partially addressed using a fat pad reconstitution experiment that demonstrates the dependence of ICAM-1 positive luminal progenitor cells on HGF stimulation. This one in vivo experiment is potentially very important and should be expanded upon to show some functional role of ICAM-1 in mammary gland development (Figure 3–figure supplement 5). Firstly, the number of experimental mice should be increased to obtain a meaningful statistical result. These results should then be incorporated into a main figure. Moreover, some IHC analysis to show the localisation of ICAM-1 luminal progenitor cells during mammary gland development would be useful*.

Fat pad reconstitution experiments:

As requested by the reviewers, we have performed additional transplantation experiments with untreated and HGF-treated luminal progenitor cultures. The data are shown in a main figure (Figure 7) and in Figure 7–figure supplement 1. They are described in a specific paragraph of the Results section (subsection “HGF stimulation confers regenerative potential on luminal progenitors”). The detailed protocol is presented in Materials and methods (“Transplantation assays and whole mount analysis”).

To complement the in vivo data previously obtained with cells isolated from primary spheres (ex Figure 3–figure supplement 5), we transplanted secondary spheres harvested from untreated and HGF-stimulated cultures of luminal progenitors. As described in Figure 6 (ex Figure 5), HGF-treated secondary spheres, when compared to untreated cultures, are greatly enriched in K5-positive cells (20% vs <1%). As expected, we found that only HGF-treated secondary spheres displayed regenerative potential (Figure 7) and developed ductal-type outgrowths comprising basal myoepithelial and luminal cells (Figure 7).

Localisation of ICAM-1 luminal progenitor cells:

As suggested by the reviewers, we have studied ICAM-1 distribution in the developing mammary epithelium, using immunohistochemical approach. The data are documented in a novel main figure (Figure 9). We have introduced a new paragraph in the Results section (subsection headed “ICAM-1 can serve to localize luminal progenitors in the mammary epithelium”) and implemented the Material and methods accordingly (in the subsection headed “Immunofluorescence and immunohistochemical stainings”).

The ICAM-1 distribution pattern in the mammary epithelium of pubertal and late pregnant females, as revealed by immunohistochemical staining (Figure 9), was in agreement with the flow cytometry data shown in Figure 1. In addition, we found that ICAM-1 expression was down-regulated at the onset of lactation.

We observed that in virgin mature mice, ICAM1-positive cells formed clusters within the ducts, a property consistent with their clonogenic potential (Figure 9). At early pregnancy, luminal cells located in large ducts were negative for ICAM-1 (Figure 9). In contrast, in the developing alveoli and small ducts, the luminal layer, presumably enriched in alveolar progenitors, contained clusters of cells positive for ICAM-1 (Figure 9). In agreement with the qPCR data, ER-positive cells were mostly ICAM1-negative (Figure 9).

In the mammary epithelium of mature virgin and early-pregnant mice (Figure 9), ICAM-1 expression was restricted to cell-cell contacts, basal-to-basal, basal-to-luminal and luminal-to luminal.

We have added a sentence in the Discussion (at the end of the subsection headed “ICAM-1 is a new marker for the enrichment of clonogenic luminal progenitors”) mentioning that ICAM-1 can be used to localize and further characterize luminal progenitors in situ.

*2) Previous work published by*
[19]
*has demonstrated the role of MET and HGF in the luminal progenitor cell function in vivo, albeit not specifically for the ICAM-1 positive subpopulation. Since the novelty of this manuscript relies on the identification of ICAM-1 as a marker of luminal progenitor cells and as the generation of basal-like cells is dependent on c-Met, it is important to consolidate the functional contribution of ICAM-1 to this process. Is ICAM-1 necessary for EMT/the transition of progenitor cells to basal like cells in response to HGF? Furthermore, based on the knowledge that ICAM-1 is a co-receptor of c-Met (reported in an independent publication by one of the co-authors), the novelty of the report could be substantially expanded by the assessment whether ICAM-1 also functionally contributes to the basal cell differentiation process. In vivo experiments addressing directly the requirement for ICAM-1 are therefore necessary to address this point*.

Using *Icam1*-KO mice, we analyzed the functional contribution of ICAM-1 in mammary development and in the luminal progenitor response to HGF. The results of these experiments are presented in Figure 9—figure supplement 1 and described in an additional paragraph at the end of the Results section (in the subsection “Loss of ICAM-1 does not impede the response of luminal progenitors to HGF”).

Whole mount and flow cytometry analysis performed in adult virgin and pregnant females showed that loss of ICAM-1 neither visibly affected mammary morphogenesis, nor perturbed the balance between basal myoepithelial and luminal cell populations (Figure 9—figure supplement 1).

To more specifically investigate the functional contribution of ICAM-1 to the luminal progenitor response to HGF, we isolated Sca1-negative luminal progenitors from *Icam1*-deficient mammary glands and treated them with HGF. As reported for ICAM1-expressing luminal progenitors, HGF was able to stimulate the clonogenic activity of *Icam1*-deficient luminal progenitors and modulate the expression of lineage- and EMT- specific genes (Figure 9—figure supplement 1).

Altogether, these experiments show that ICAM-1 is not mandatory for mammary development, HGF/Met signaling in luminal progenitors and basal cell differentiation process, and suggest that CD44v6 is the main co-receptor for Met in the mammary epithelium. The Discussion part has been modified according to these conclusions (please see the subsection “ICAM-1 is not mandatory for HGF/Met signaling in luminal progenitors”).

*3) There are also some specific ambiguities with regard to*
Figure 4
*and*
Figure 4*, specifically in the E-cadherin staining. In*
Figure 4*, E-cadherin-mediated cell-cell adhesion does not seem to change with and without HGF-stimulation? In*
Figure 4*, the absence of surface E-cadherin staining in K5-positive cells is not obvious. Better resolution microscopy images should be provided. Claudin-I immunofluorescence staining may resolve the unclear picture of the maintenance/loss of cell adhesion. It would also be helpful to validate the EMT-like changes in gene expression on the protein levels for example using immunofluorescence imaging*.

E-cadherin-mediated cell-cell adhesion/ Claudin-I immunofluorescence staining:

An additional main figure has been created (Figure 5) to better document cell-cell adhesion in spheres derived from luminal progenitors. Panels A and B are dedicated to E-cadherin expression, panel C to claudin-1 detection.

As previously shown (ex Figure 4), E-cadherin-mediated cell-cell adhesion was not dramatically affected in HGF-stimulated spheres (Figure 5). To underscore this, we have replaced the panel showing double immunolabeling for K5 and E-cadherin (ex Figure 4) by microphotographs documenting Snail2 and E-Cadherin expression (Figure 5), and modified the text in the Results section (in the third paragraph of the subsection headed “Luminal progenitors activate EMT program and repress luminal-specific regulatory genes upon stimulation by HGF”).

Using double immunofluorescence labeling for E-Cadherin and claudin-1, we found that in untreated spheres, most cells expressed claudin-1 at their junctions, whereas HGF-treated spheres clearly displayed large cell areas lacking claudin-1 (Figure 5). These data are in agreement with the important down-regulation of *Cldn1* observed by qPCR (Figure 4). Comments related to Claudin-1 expression have been added in Results section.

Validation of the EMT-like changes in gene expression on the protein levels:

As suggested by the reviewers, we have analyzed p63 and Snail2 expression in untreated and HGF-treated primary spheres using immunofluorescence labeling. The images are shown in Figure 4, and related comments have been added in the Results section. The Materials and methods section has been revised to introduce the corresponding protocol.

In agreement with the gene expression analysis, we detected numerous cells with a nuclear staining of p63 and Snail2 in HGF-treated spheres, whereas untreated cells were negative for these markers.

*4) Several experiments lack statistical analysis and these should be addressed to strengthen the conclusions being made*.

We have increased the number of experiments for the data presented in the main Figures 6 and 7 of the revised manuscript.

Additionally, as requested, we have provided source data files corresponding to each main figure. It is important to mention that each cell sorting experiment presented in the manuscript was performed with mammary epithelial cells isolated from pooled glands of, at least, 3 mice (cf. source data files). ICAM-1 distribution in pubertal and late-pregnant mice were not extensively analyzed, since at these stages, almost all the luminal compartment is either negative or positive for ICAM-1 (Figures 1 and 8).